# On the origin of species thermodynamics and the black hole - tower correspondence

Alvaro Herráez[1], Dieter Lüst[1,2], Joaquin Masias[1] and Marco Scalisi[1,3,4]

**1** Max-Planck-Institut für Physik (Werner-Heisenberg-Institut),
Boltzmannstr. 8, 85748 Garching, Germany
**2** Arnold Sommerfeld Center for Theoretical Physics,
Ludwig-Maximilians-Universität München,
80333 München, Germany
**3** Department of Physics and Astronomy "Ettore Majorana",
University of Catania, Via S. Sofia 64, I-95125 Catania, Italy
**4** INFN - Sezione di Catania, Via S. Sofia 64, I-95123 Catania, Italy

## Abstract

Species thermodynamics has been proposed in analogy to black hole thermodynamics. The entropy scales like an area and is given by the mere counting of the number of the species. In this work, we *derive* the constitutive relations of species thermodynamics and explain how those *originate* from standard thermodynamics. We consider configurations of species in thermal equilibrium inside a box of size $L$, and show that the temperature $T$ of the system, which plays a crucial role, is always upper bounded above by the species scale $\Lambda_{sp}$. We highlight three relevant regimes: (i) when $L^{-1} < T < \Lambda_{sp}$, and gravitational collapse is avoided, the system exhibits standard thermodynamics features, for example, with the entropy scaling like the volume of the box; (ii) in the limit $L^{-1} \simeq T \to \Lambda_{sp}$ we recover the rules of species thermodynamics with the entropy scaling like the area of the box; (iii) an intermediate regime with $L^{-1} \simeq T < \Lambda_{sp}$ that avoids gravitational collapse and fulfills the Covariant Entropy Bound; this interpolates between the previous two regimes and its entropy is given simply in terms of the counting of the species contributing to the thermodynamic ensemble. This study also allows us to find a novel and independent bottom-up rationale for the Emergent String Conjecture. Finally, we present the *Black Hole - Tower Correspondence* as a generalization of the celebrated Black Hole - String Correspondence. This provides us with a robust framework to interpret the results of our thermodynamic investigation. Moreover, it allows us to qualitatively account for the entropy of black holes in terms of the degrees of freedom of the weakly coupled species in the tower.

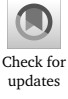

# 1   Introduction

Understanding the generic properties of Effective Field Theories (EFTs) arising from Quantum Gravity (QG) is at the core of trying to connect the latter with the observable universe. In the context of the Swampland Program [1], these generic properties are usually formulated in terms of conjectures, which can be examined and, sometimes, rigorously formulated in concrete frameworks, such as different corners of String Theory or AdS/CFT (see also [2–8] for recent reviews on the topic). Interestingly, such explorations usually provide insights into a wide range of topics related to QG, from constraints in the spectrum of particles or energies allowed in the resulting EFTs, to black hole physics, and every so often these turn out to be related in unexpected ways.

One of the best established properties of QG theories is the existence of infinite towers of states in the spectrum. This is not necessarily the case in standard QFT, namely when gravity can be neglected. Furthermore, according to the Distance Conjecture [9], one of the best established Swampland Conjectures, some of these infinite towers always become massless (in $d$-dimensional Planck units for a $d$-dimensional EFT) as one approaches infinite distance limits in moduli (or, more generally, in scalar field) space. In fact, the Emergent String Conjecture [10] constraints the nature of these light towers by postulating that they can only be of

two kinds: either Kaluza-Klein (KK)-like towers, or towers of oscillator modes from a weakly coupled critical string. In this sense, the presence of towers of states becoming light indicates the existence of a finite range of validity (in field space) for such EFTs (see e.g. section II.B of [11]). One could think that by integrating in the different states as they become light could be enough to define a new EFT with an extended range of validity. However, the fact that these towers include *infinitely* many degrees of freedom (dof) hints towards the existence of a scale above which a usual EFT description cannot be applied. This can be seen as a maximum cutoff for *any* gravitational EFT, and in the limit of such infinite tower of states becoming massless, this fundamental cutoff must also vanish (in $d$-dimensional Planck units).

This relevant cutoff scale is the so-called Species Scale [12–15], originally defined as

$$\Lambda_{\text{sp}} \simeq \frac{M_{\text{Pl},d}}{N_{\text{sp}}^{\frac{1}{d-2}}}\,, \tag{1}$$

where $M_{\text{Pl},d}$ is the $d$-dimensional Planck mass and $N_{\text{sp}}$ counts the number of dof with masses below the species scale itself. Notably, this scale encapsulates the idea that a gravitational EFT with an arbitrarily high number of light dof breaks down at scales arbitrarily lower than $M_{\text{Pl},d}$, contrary to the naive expectation of the Planck mass giving the cutoff for gravitational EFTs. Different perturbative and non-perturbative arguments lead to the result that $\Lambda_{\text{sp}}$ gives indeed the maximum scale up to which a gravitational EFT can be trusted. In this context, in the cases in which $N_{\text{sp}}$ is dominated by the towers of states coming from KK or string oscillator modes, $\Lambda_{\text{sp}}$ reduces to the higher dimensional Planck mass or the string scale, respectively, as one would expect (see e.g. [16–19]). More recently, this idea of considering the species scale as the maximum cutoff scale for gravitational EFTs was formulated, using a more standard EFT language, as the scale suppressing higher curvature corrections [18], schematically as

$$S_{\text{EFT},d} = \int d^d x \sqrt{-g} \, \frac{M_{\text{Pl},d}^{d-2}}{2} \left( R + \sum_n \frac{\mathcal{O}_n(R)}{\Lambda_{\text{sp}}^{n-2}} \right) + \dots\,, \tag{2}$$

which can be more easily extended to the interior of moduli spaces. In [18], and the subsequent work [20–24], it was indeed confirmed that the species scale computed from protected higher curvature corrections in different supersymmetric setups indeed matches the expected results, recovering asymptotically the string scale or the higher-dimensional Planck scale depending on whether the limits that are probed are emergent string limits or decompactification limits, respectively.

A complementary derivation of the species scale comes from black hole physics [12,13,20]. In this case, $\Lambda_{\text{sp}}$ is given as the scale associated to the smallest semi-classical black hole in a $d$-dimensional gravitational EFT. This notion of species scale is compatible with, and deeply related to, the aforementioned definitions, since corrections to the EFT also affect the size of the horizon of the black holes and can become particularly relevant for small black holes. It also connects with one of the most interesting questions in QG, namely the explanation for the entropy of black holes. In particular, a Schwarzschild black hole of size $\Lambda_{\text{sp}}^{-1}$ has an entropy given by

$$S \sim \left( \frac{M_{\text{Pl},d}}{\Lambda_{\text{sp}}} \right)^{d-2} \simeq N_{\text{sp}}\,. \tag{3}$$

Note that this entropy does not just follow the usual area law but, in this case, it also turns out to be given by the total number of species. In fact, the universality of the area law for the entropy of species was argued for in [25] from the saturation of unitarity. Besides, the observation that a minimal black hole corresponds to a particular tower of states led to the formulation of *species thermodynamics* [26], which describe the statistics of towers of states

upon identifying the entropy with the number of species, and the energy of the system with the mass of the minimal black hole. Using then the laws of black hole thermodynamics, the corresponding laws of species thermodynamics were deduced (see section 3.4). In [27], it was also shown how the equivalence between minimal black holes and species can strongly constrain the mass spectrum and the degeneracies of the species particles. This result was in agreement with the emergent string conjecture [10] and, therefore, constituted a first bottom-up argument for it.

In this work, we shed light on the origin of the peculiar properties of species thermodynamics. We find the following key results:

- Using *standard thermodynamics* for a system of species in thermal equilibrium inside a box of size $L$, we *derive* the constitutive relations of species thermodynamics for the entropy and the energy, i.e. $S \sim N_{sp}$ and $E \sim N_{sp} \Lambda_{sp}$, in the limit $T \to \Lambda_{sp}$. This naturally produces the scaling of the entropy with the area in such limit.

- We recover the standard volume dependence of $S$ and $E$ in the regime $L^{-1} < T < \Lambda_{sp}$, and show that a box of species effectively interpolates between the two regimes while still avoiding gravitational collapse and fulfilling the Covariant Entropy Bound.

- We show the existence of a *maximum temperature*, which can be at most equal to $\Lambda_{sp}$. This result is directly derived by the application of entropy bounds to a finite box of species.

- We show that the only *appropriate*, i.e. towers consistent with the properties of species thermodynamics, are those with polynomial or exponential degeneracies, which is reminiscent of KK and string towers, respectively. This provides a novel and independent bottom-up rationale for the emergent string conjecture (other indications are offered in [27–29]).

- We present the *Black Hole - Tower Correspondence* by generalizing the original argument leading to the celebrated the Black Hole - String correspondence [30] (see also [31,32]). We show that the correspondence is, in fact, between black holes and the aforementioned allowed towers of states, namely either KK or string modes.

We start our investigation by considering the thermodynamics of species in a box of size $L$ at a temperature $T$, in the canonical ensemble. It is a known result that this entropy generically scales like the volume of the box [33,34]. This may look like an important obstacle to recover the right dependence of the entropy on the area as dictated by species thermodynamics (see also the area dependence of the entropy of species [25]). However, a first hint that this might not be the whole story is the fact that, in the presence of gravity, the maximum entropy of a system is constrained by the Covariant Entropy Bound (CEB) [35,36] and, in the case of the box, this means it is bounded above by its area (in Planck units). Moreover, there is always the possibility that if the box with the species is heated up too much, it could collapse to a black hole. We show that, as we increase the temperature and decrease the length $L$ of such a box, gravitational collapse is avoided and the CEB fulfilled if the temperatures stay below $\Lambda_{sp}$ and and the box is taken sufficiently small. It is precisely in the limit $T \simeq 1/L \simeq \Lambda_{sp}$ that the CEB is saturated, gravitational collapse should take place, and also $S \to N_{sp}$ [37]. In this limit, the entropy turns out to be proportional to the area of the box in which the species are contained.

Moving away from the limit in which $T \simeq \Lambda_{sp}$, namely for temperatures lower than the species scale, we show that the entropy of the system cannot be given in terms of the area of the box. There are two relevant situations. On the one hand, when a small number of species is involved, we recover the standard result of the entropy dependence on the volume of the box. On the other hand, we can consider high enough temperatures to allow for more species

to contribute to the ensemble. However, in this case, one must avoid an increase of entropy such that the system collapses into a black hole or violates the CEB. This is possible if $T \simeq 1/L$, even for temperatures below $\Lambda_{\text{sp}}$. In this case, we obtain a dependence of the entropy like

$$S \sim N_T \,, \tag{4}$$

where $N_T$ is the effective number of species at temperature $T$. Importantly, this converges to $N_{\text{sp}}$ as $T$ approaches $\Lambda_{\text{sp}}$. This case, therefore, effectively interpolates between the volume and area laws for the entropy in the field theory regime.

We would like to emphasize that we perform our whole analysis in the limit in which we can neglect the $d$-dimensional gravitational interactions, namely for $M_{\text{Pl},d} \gg \Lambda_{\text{sp}}$, or equivalently $N_{\text{sp}} \gg 1$. Such a limit would correspond to asymptotic limits of moduli space in a consistent string effective description.

This effective thermodynamic analysis allows us also to select consistent type of towers. Specifically, from the convergence of the EFT canonical partition function, we obtain that the degeneracy cannot be superexponential that is, it cannot be larger than the degeneracy of string oscillators. Moreover, from the requirement that $S \sim N_T$ converges to $S \sim N_{\text{sp}}$ when $T$ approaches $\Lambda_{\text{sp}}$, we obtain that the tower degeneracy has to be at least polynomial, which corresponds to KK-like towers. Thus, we are able to exclude other types of degeneracy and find a novel supporting bottom-up argument for the Emergent String Conjecture [10]. Our results complement recent evidence in the context of species thermodynamics [27, 28] and gravitational scattering amplitudes [29].

One of the main lessons is therefore that a finite box of species, with mass scale degeneracy as the one of KK or string modes and at finite temperature $T$, turns out to have the peculiar properties of species thermodynamics when $T \to \Lambda_{\text{sp}}$. In this limit, the system would collapse into the smallest possible black hole with temperature $T \sim \Lambda_{\text{sp}}$ and entropy $S \sim N_{\text{sp}}$.

As a further non-trivial step, one could imagine increasing the size of such small black hole, effectively making it more and more semi-classical, while keeping its entropy constant, by varying the ratio $\Lambda_{\text{sp}}/M_{\text{Pl},d}$. This represents a first concrete example of a correspondence between tower of species in the arbitrarily weakly coupled EFT regime and gravitational strongly coupled objects, such as large black holes.

This takes us to the final part of our investigation, where we propose and investigate this *black hole - tower correspondence*. We show that this is a generalization of the black hole - string correspondence, as originally proposed in [30]. We would like to note that a correspondence between minimal black holes and towers was already explored in [27]. Our proposal extends beyond that and allows to set a correspondence between black holes of any size and towers of species at arbitrarily weak gravitational coupling, all the way along constant entropy lines. It provides us with an effective tool to qualitatively account for the entropy of black holes in terms of the degrees of freedom of the species in the tower.

We consider varying a modulus in order to control the effective gravitational coupling of $d$-dimensional gravity (i.e. the quotient $\Lambda_{\text{sp}}/M_{\text{Pl},d}$). We then draw the correspondence between black hole solutions (when gravity is sufficiently strong), and a box of species arbitrarily weakly coupled in the opposite regime (see Fig. 5). On the black hole side, the entropy goes like $S \sim (R_{\text{BH}}/M_{\text{Pl},d})^{d-2}$, whereas in the EFT regime it interpolates between the volume dependence (when few species are included) and $S \sim N_T$ when the species dominate. We prove that a correspondence between these two entropies and energies takes place precisely when both limits approach $T \sim \Lambda_{\text{sp}}$ and $S \sim N_{\text{sp}}$. Similar arguments focusing precisely on this correspondence point, like the ones presented in [27, 28], can be understood in our picture as the requirement for the species entropy and species energy to be the *appropriate* (see section 3.3) from the EFT point of view to allow for the black hole - tower correspondence to take place. We show this can only happen for towers of states with polynomial or exponential degeneracy, consistent with arguments given in [27, 28] and [29]. This offers a complete picture

in which the EFT results on the one side match the black hole results on the other side if the correspondence between the black hole and the ensemble occurs. In fact, in the spirit of the original reasoning behind the black hole - string correspondence, proposed in [30], one can consider semi-classical black holes with a very large horizon and follow their constant entropy lines as the modulus is varied. In such a way, one can get to the point where the entropy of the original black hole is accounted for by the degrees of freedom of the species in the tower, namely the free string (as originally proposed) or the KK tower (as would be the case if the original black holes wrapped extra dimensions).

The structure of this paper is as follows. In section 2, we review how the CEB and gravitational collapse constraint thermal configurations that can be described in field theory, highlighting the importance of the limit $\Lambda_{\mathrm{sp}} \simeq T \simeq 1/L$, in which both bounds are saturated and one recovers $S \sim N_{\mathrm{sp}}$. In section 3, we study the EFT thermodynamics of a system of species at temperature $T \le \Lambda_{\mathrm{sp}}$ (mainly in the canonical ensemble) in the limit in which interactions can be neglected. We also obtain the usual volume dependence, as well as the dependence on the effective number of degrees of freedom at temperature $T$, for the entropy in the relevant limits, finding EFT evidence for species thermodynamics. Furthermore, we review the agreement of our results with previous results by treating this limiting case in the microcanonical ensemble. Along the way, we explain how convergence of the canonical partition function, together with the limit $N_T \to N_{\mathrm{sp}}$ as $T \to \Lambda_{\mathrm{sp}}$ constraint the possible towers to those with polynomial or exponential degeneracy. In section 4, we revisit our previous analysis from the top-down embedding of the towers with polynomial degeneracy, comparing black hole and black brane solutions in the presence of extra dimensions. Finally, in section 5, we review the black hole - string correspondence and heuristically extend it to a black hole - tower correspondence, presenting the bigger picture of how the species entropy can be understood as the intermediate step to account for the entropy of a general Schwarzschild black hole from the EFT counting of the entropy of a box of weakly coupled species in the appropriate limit. We leave a summary of our findings and conclusions for section 6, and relegate some technical details to the appendices.

## 2 IR/UV mixing and the covariant entropy bound

We begin by reviewing the Covariant Entropy Bound (CEB) [35,36], focusing on the IR/UV mixing that arises when considered in the context of Effective Field Theories (EFTs) of gravity. The basic idea that we will exploit is the fact that the maximum entropy that can be accommodated in a finite spacelike region, $\Sigma$, behaves holographically in the presence of gravity. That is, it is bounded by the area of the boundary of said region, measured in Planck units,[1]

$$S(\Sigma) \le \frac{A(\partial \Sigma)}{4\,\ell_{\mathrm{Pl,d}}^{d-2}}\,, \tag{5}$$

where $A(\partial \Sigma)$ refers to the area of the boundary of $\Sigma$, and $\ell_{\mathrm{Pl,d}}$ to the Planck length in $d$-dimensions.[2] From confronting this area behaviour with the volume scaling of the entropy in field theory, which for $N$ species of particles in the thermodynamic limit, characterized by the temperature $T$, takes the form

$$S_{\mathrm{EFT}}(T, \Sigma) \sim N\,T^{d-1}\mathrm{vol}(\Sigma)\,, \tag{6}$$

---

[1]To be precise, in order to apply the spacelike projection theorem to obtain the simple form of the bound applied to a spacelike region, $\Sigma$, we need the region to be contained in the causal past of the future directed light-sheets (i.e. the future-directed ligthsheets are complete) [35,36].

[2]We use the conventions in which the Planck length is related to the $d$-dimensional Newton constant and (reduced) Planck mass in the following way $G_{\mathrm{N},d} = \ell_{\mathrm{Pl,d}}^{d-2} = 1/8\pi M_{\mathrm{Pl},d}^{d-2}$.

it is easy to see that the IR/UV mixing arises from the fact that probing larger and larger regions restricts the maximum temperature that one can probe without violating the bound. In particular, if we consider the region $\Sigma$ to be characterized by the length scale $L$ (e.g. by taking a $d$-dimensional sphere of radius $L$), we obtain the following bound

$$N \left( T \, \ell_{\mathrm{Pl,d}} \right)^{d-1} \lesssim \left( \frac{\ell_{\mathrm{Pl,d}}}{L} \right), \tag{7}$$

which very neatly displays the aforementioned IR/UV mixing between the IR scale of the configuration under consideration, $L$, and the UV scale, $T$.

The logic yielding constraints on low energy EFTs of gravity coming from direct application of the CEB is the following. First, we note that *every* configuration in the theory must fulfill the bounds (5)-(7). Thus, if one finds a configuration that naively violates the bound, two options arise. Either that particular configuration turns out to be censored in the EFT via some mechanism in (quantum) gravity precluding it, or the regime of validity of the EFT must be restricted to avoid that configuration. Whenever a particular configuration can be argued to be within the regime of validity of a low energy EFT, one would expect the latter option to be the correct one. In particular, a very neat implementation of this idea is to restrict the regime of validity of the EFT via bounding the maximum UV cutoff that can be considered. This reasoning was originally used in [37] in the context of families of AdS vacua to bound the species scale [12–15], or equivalently the mass scale of a tower, in terms of the cosmological constant, as originally predicted by the AdS Distance Conjecture [38].[3] Furthermore, it has also been used in combination with the idea of dynamical cobordisms [42–51], to provide a bottom-up rationale for the Distance Conjecture [52].[4] In these works, the configurations whose entropy was confronted with the CEB had two important features. First, the maximum energy up to which the EFT was assumed to be trustable, was identified with the species scale. Second, the entropy that was confronted with the CEB in order to obtain the bounds on such UV cutoff of the EFT (i.e. the species scale) was the one associated to configurations in which only the massless subsector of the EFT was considered, i.e. $N \sim 1$ in eq. (6). In this work, we consider a different type of configurations, including not only the massless subsector of the theory. In particular, we consider configurations in thermal equilibrium at temperature $T$, with special emphasis in the limit $T \to \Lambda_{\mathrm{sp}}$, and study the thermodynamics of all the species which contribute significantly to such configurations, which we denote $N_T$.

At this point it is natural to wonder which configurations are then the right ones to consider in order to obtain useful constraints for gravitational EFTs. The answer lies on the aforementioned fact that *every* configuration must fulfill the CEB. Thus, the bounds and results from our analysis here are independent and complementary to the previous results obtained by applying the same logic to different kinds of configurations.

To begin with our exploration, let us consider a spherical region characterized by a typical size $L$, and study configurations in thermodynamic equilibrium at temperature $T$ that fit in it, including the potential contributions from all possible species, whose number is given by $N_T$ and generally depends in the temperature (and the density of one-particle states). In particular, we want to focus on the case in which a tower of massive particles is present, and thus the species scale, which gives an upper bound for the EFT cutoff, can be well described by

$$\Lambda_{\mathrm{sp}} \simeq \frac{M_{\mathrm{Pl},d}}{N_{\mathrm{sp}}^{\frac{1}{d-2}}}. \tag{8}$$

---

[3]See also [11,39–41] for some recent discussions on the species scale in the context of cosmology.

[4]See also [53–55] for previous bottom-up arguments recovering also some of the key features of the Distance Conjecture from different approaches.

For concreteness, let us now consider the following parameterization for the mass spectrum of the aforementioned towers at level $n$ [37] (we will elaborate more on the possible types of towers and parameterizations in section 3, but for now we stick to this one for concreteness, and keep in mind that the subsequent discussion can be adapted to more general cases)

$$m_n = n^{1/p} m_{\mathrm{t}}. \tag{9}$$

Here, $m_{\mathrm{t}}$ refers to the mass scale of the tower and $p$ is an effective way to encode the density of the tower, which captures the behaviour of towers of KK modes from $p$ compact dimensions (for finite values of $p$), and also the key features of a tower coming from weakly coupled string oscillators (in the $p \to \infty$ limit). According to the Emergent String Conjecture [10], these are the only two relevant types of towers as one approaches infinite distance limits in moduli space, and recent results support this claim from a bottom-up approach [27–29]. In the presence of a tower with mass spectrum given by (9), one can then identify the maximum $n$ that contributes to the species scale with the number of species, $N_{\mathrm{sp}}$, and thus obtain

$$N_{\mathrm{sp}} \simeq \left( \frac{\Lambda_{\mathrm{sp}}}{m_{\mathrm{t}}} \right)^p . \tag{10}$$

Thus, for a temperature $T \leq \Lambda_{\mathrm{sp}}$, we can similarly define the number of species with masses below $T$, which turn out to be the ones that give the leading contribution to the thermodynamic quantities, as

$$N_T \simeq \left( \frac{T}{m_{\mathrm{t}}} \right)^p . \tag{11}$$

We can then use eqs. (10) and (8) to rewrite $\Lambda_{\mathrm{sp}}$ in terms of the mass scale of the tower, $m_{\mathrm{t}}$, and the density parameter, $p$, and use that to obtain an expression of $N_T$ in terms of the temperature and the species scale

$$N_T \simeq \left( \frac{T}{\Lambda_{\mathrm{sp}}} \right)^p \left( \frac{M_{\mathrm{Pl},d}}{\Lambda_{\mathrm{sp}}} \right)^{d-2} \simeq \left( \frac{T}{\Lambda_{\mathrm{sp}}} \right)^p N_{\mathrm{sp}} , \tag{12}$$

showing that $N_T \leq N_{\mathrm{sp}}$ as long as $T \leq \Lambda_{\mathrm{sp}}$. By applying the CEB in the form given by (7), with $N = N_T$, we thus obtain

$$T \lesssim \left( \frac{\Lambda_{\mathrm{IR}}}{\Lambda_{\mathrm{sp}}} \right)^{\frac{1}{d-1+p}} \Lambda_{\mathrm{sp}} \equiv T_{\max} \leq \Lambda_{\mathrm{sp}} , \tag{13}$$

where we defined $\Lambda_{\mathrm{IR}} \equiv 1/L$. From here, it is manifest that for configurations in which all the species at thermodynamic equilibrium are considered, there is a maximum temperature which is compatible with the CEB, and it is, in general, strictly lower than the species scale. However, there is a particularly interesting case in which this temperature can be pushed up to the species scale, namely when we study configurations for which $\Lambda_{\mathrm{IR}} \simeq \Lambda_{\mathrm{sp}}$, since then $T_{\max} \simeq \Lambda_{\mathrm{sp}}$ and, obviously, $N_T \simeq N_{\mathrm{sp}}$. This limiting case, originally highlighted in [37], corresponds to the limit in which the entropy of the thermodynamic configuration would contain the same entropy as the minimal black hole in the EFT (i.e. the one with the smallest possible radius, $R_{\mathrm{BH}} \simeq \ell_{\mathrm{sp}}$),[5] and thus approaches the maximum entropy allowed in such a region. Equivalently the box of species at temperature $T \simeq 1/L \to \Lambda_{\mathrm{sp}}$ approaches its maximum possible temperature. This entropy goes like

$$S \sim N_{\mathrm{sp}} \simeq \left( \frac{M_{\mathrm{Pl},d}}{\Lambda_{\mathrm{sp}}} \right)^{d-2} , \tag{14}$$

---

[5]These ideas regarding the maximum temperature, or equivalently the minimum length, of a (semi-classical) black hole were in fact used as one of the original non-perturbative motivations for the species scale in [13].

recovering the area behaviour of the black hole entropy from species counting. Similarly, the smallest black holes in the EFT were the main focus of [26], where the entropy associated to the species was actually promoted to a fundamental concept, and the corresponding laws of *species thermodynamics* were proposed. One of the main goals of this paper is to *explain* this dependence of the entropy from basic thermodynamic considerations and comparison with the CEB.

There is another remarkable feature of the limit $T \simeq 1/L \to \Lambda_{\rm sp}$, which was also noticed in [37], namely the fact that once it is approached, the CEB actually coincides with the bound from preventing gravitational collapse (à la CKN [56]). The latter is in general stronger, since a generic configuration in the EFT satisfying $T \simeq T_{\rm max}$, as given in (13), will have a Schwarzchild radius larger that $L = \Lambda_{\rm IR}^{-1}$. To see this, recall that the ADM mass of the black hole is related to its Schwarzschild radius in $d$-dimensions as

$$R_{\rm BH} \simeq \left( \frac{M_{\rm BH}}{M_{{\rm Pl},d}} \right)^{\frac{1}{d-3}} M_{{\rm Pl},d}^{-1} \,, \tag{15}$$

and that the total energy inside the box of size $L$, which we can identify with the ADM mass of the configuration, reads

$$E = N_T L^{d-1} T^d \,. \tag{16}$$

Requiring that such configuration does not collapse into a black hole then boils down to requiring

$$R_{\rm BH} \leq L \,. \tag{17}$$

This leads to a bound on the temperature

$$T \leq T_{\rm CKN} = \left( \frac{\Lambda_{\rm IR}}{\Lambda_{\rm sp}} \right)^{\frac{2}{d+p}} \Lambda_{\rm sp} \leq T_{\rm max} \,, \tag{18}$$

which is strictly lower than the bound found using the CEB for any $d \geq 2, p \geq 1$, for any $L > \Lambda_{\rm sp}^{-1}$. As announced, the two bounds match when $L^{-1} \simeq T \to \Lambda_{\rm sp}$. This is particularly interesting since, even though the bounds from gravitational collapse give similar qualitative results than the ones obtain from applying the CEB, some of the quantitative outputs can be slightly different. In this sense, all the arguments and considerations that specifically refer to the limiting case above are particularly robust against any subtleties related to gravitational collapse.

Having discussed both the CEB and the gravitational collapse bound, and given that the latter is always strictly more constraining than the former, a natural question arises: Can we approach the maximum entropy configuration (i.e. the saturation of the CEB) with a thermal ensemble that has not collapsed gravitationally? The answer, as can be seen from the previous analysis, turns out to be positive, since for any configuration with $T \simeq 1/L$, both bounds are satisfied as long as $T < \Lambda_{\rm sp}$, and in the limit in which $T \to \Lambda_{\rm sp}$ we have seen that both bounds are saturated and we expect our configuration to collapse to a black hole without a dramatic increase on its entropy, since it is already parametrically the same as that of the corresponding black hole solution with the same size.

## The observable universe and extra dimensions

Let us now make a small detour and consider some potential implications of our previous discussion when boldly applied to the observable universe. In the presence of KK towers, the states in the tower contribute to the entropy if $T \gtrsim m_{\rm t} = 1/r$. Imposing this on $T_{\rm max}$ in eq. (13)

leads to a mixing between IR and UV scales

$$\frac{\Lambda_{\text{IR}}}{M_{\text{Pl},d}} > \left(\frac{\Lambda_{\text{sp}}}{M_{\text{Pl},d}}\right)^{\frac{(d-1)(d-2+p)}{p}}, \tag{19}$$

or equivalently, in terms of the size of the compact dimensions

$$\frac{\Lambda_{\text{IR}}}{M_{\text{Pl},d}} > \left(\frac{1/r}{M_{\text{Pl},d}}\right)^{d-1}. \tag{20}$$

For a box the size of the observable universe, we have $\Lambda_{\text{IR}} = \sqrt{\Lambda_{cc}}$, which bounds the size of the extra dimensions as

$$r > \left(\frac{\Lambda_{cc}}{M_{\text{Pl},d}^2}\right)^{\frac{1}{2(d-1)}} \ell_{\text{Pl,d}}. \tag{21}$$

For $\Lambda_{cc} \simeq 10^{-120} M_{\text{Pl}}^2$ we find

$$r > 10^{-15}\,m. \tag{22}$$

For $T_{\text{max}} < 1/r$ the EFT would be oblivious to the existence of species (or equivalently, to the presence of extra dimensions). The cutoff temperature $T = T_{\text{max}}$ may not signal out a breakdown of the EFT if the effective description changes at some lower temperature, due to, for example, the presence of the extra dimensions or the back-reaction of the boundary.

## 3 Tower entropy from EFT thermodynamics

In this section we consider the thermodynamics of a set of different species of particles in a box. The main goal is to begin by reviewing some standard results about the entropy of such configurations, and how it relates to the temperature, the size of the box, and the number of particles, in different limits that will be of particular interest later in this work.

### 3.1 Canonical ensemble

We start by considering a system with energy levels labeled $E_r$, where the energy is given by a sum of the energies of each of the constituents in the system. This boils down to assuming that we can neglect the contribution to the energy coming from the gravitational interactions between the particles, and in the systems in which we focus on along this work it is motivated by the fact they lie near infinite distance limits in moduli space, where $d$-dimensional gravity is arbitrarily weakly coupled (we will elaborate more on this along the way). Thus, we can write $E_r = \sum_{\{\kappa_r\}} E_{1,\kappa_r}$, where $\{\kappa_r\}$ encode all the information about the one-particle states that constitute the system, with energy denoted by $E_{1,\kappa_r}$, like momentum distribution, mass degeneracy, etc. The canonical partition function for such a system can be written as a sum over (multiparticle) energy levels[6]

$$\mathcal{Z} = \sum_r e^{-E_r/T} = \sum_r \prod_{\{\kappa_r\}} e^{-E_{1,\kappa_r}/T}. \tag{23}$$

The product over the $\{\kappa_r\}$ includes information of the mass spectrum in the partition function. For high enough degeneracies, namely for particle spectra with an exponential (or greater) degeneracy, the partition function may not be well-defined, signaling a breakdown of our

---

[6]This system can be expressed in terms of its Hamiltonian as $\mathcal{Z} = \text{Tr}\{e^{-H/T}\}$.

effective description, such that one should re-express the theory in term of more appropriate degrees of freedom. In the case in which the partition function converges for high degeneracies (e.g. exponential below some critical temperature), it will be dominated by the single-particle partition function at mass $m \simeq T$ [57], whereas for small degeneracies (e.g. polynomial or sub-polynomial) the partition function will be dominated by multi-particle configurations (i.e. with many one-particle states occupied). This has been thoroughly studied in the context of QCD [58,59], where at temperature scales near the pion mass the effective baryon description breaks down, and also in string theory, where the phase transition is in general less understood [31,32,57].

**Multi-particle state domination**

Let us then first consider a system consisting of particles of $N_T$ different species,[7] under the assumption of sufficiently small mass degeneracy, such that multi-particle states dominate the partition function. We treat the complementary case below. Each of the species is labeled by an integer $n$, and we consider configurations inside a box of size $L$ in $d$ spacetime dimensions in the canonical ensemble, namely at equilibrium at a fixed common temperature, $T$. Importantly, we have not fixed by hand the total number of species, $N_T$, nor the number of total particles in the box (equivalently, we have not fixed the number of particles of each species). Instead, we allow these numbers to fluctuate and be determined by the thermodynamics of the system, with their average values being determined as a function of $T$. Thus, one should think of the total number of particles of each species inside the box, as well as of the number of species contributing to the ensemble, as the average quantities $\langle N_n(T) \rangle$ and $\langle N_T(T) \rangle$, respectively. The (average) total number of particles in the box thus includes a sum over all species

$$\langle N_{\text{TOT}}(T) \rangle = \sum_{n=1}^{\langle N_T \rangle} \langle N_n(T) \rangle. \tag{24}$$

To avoid cluttering the notation we will usually drop the $\langle \ldots \rangle$ unless we want to emphasize the fact that this is to be thought of as an average in the equilibrium configuration at temperature $T$.

As previously mentioned, we consider the free particle limit, where the energy of the whole system can be accounted for by the energy of each of the individual constituents, neglecting interactions. The single-particle partition function for a particle of species $n$, with mass $m_n$, is given by[8]

$$\mathcal{Z}_{1,n} = \sum_{p_\alpha} e^{-E_{n,p_\alpha}/T}, \qquad E_{n,p_\alpha}^2 = m_n^2 + \sum_{\alpha=1}^{d-1} p_\alpha^2, \tag{25}$$

where $p_\alpha = k_\alpha/L$ (with $k_\alpha$ being integers) is the $(d-1)$-dimensional spatial momentum of the particles in the box. The sum over $p_\alpha$ in the partition function is over all possible momenta fitting inside the box (or equivalently over all $(d-1)$-tuples with integer entries, $k_\alpha$).

---

[7]We are choosing the notation with an eye on the fact that the total number of species contributing to the ensemble will be given by the previously introduced (and in general $T$-dependent) quantity, $N_T$, but for now we can simply consider it as a quantity to be determined and we will prove that it coincides with the number of species with masses below $T$ later in this section.

[8]We are using here the Boltzmann distribution for the (non-degenerate) one-particles states. Considering the Bose-Einstein or Fermi-Dirac distributions does not change our results in the relevant limits analyzed in this paper.

One can now build the partition function for such system of $N_T$ species, with $N_n$ identical particles of each species in the free limit, by simply multiplying the single-particle partition functions and dividing by the number of identical combinations of $N_n$ identical species

$$\mathcal{Z} = \frac{\prod_{n=1}^{N_T} \left(Z_{1,n}\right)^{N_n}}{\prod_{n=1}^{N_T} N_n!} \,. \tag{26}$$

The (average) number of particles of each species is given by the sum of all (average) occupation numbers of one-particle states (i.e. states with different $d$-dimensional spatial momenta) of said species, that is

$$\langle N_n \rangle = \sum_\alpha \langle N_{n,\alpha} \rangle = \sum_\alpha e^{-E_{n,p_\alpha}/T} \,, \tag{27}$$

where we have simply used that the average occupation number (in the absence of a chemical potential, or with a vanishing one) is given by $e^{-E/T}$ (see e.g. [33, 34]). We thus also have that $\mathcal{Z}_{1,n} = \langle N_n \rangle$

Let us now study the behaviour of $N_n$ as a function of $T$ and $L$ for different values of $m_n$ and $\Lambda_{\rm sp}$, keeping in mind that $\Lambda_{\rm sp} \geq T \geq 1/L$. Using the results of Appendix A, (c.f. eq. (A.4)) we can approximate the average occupation numbers by

$$N_n = \mathcal{Z}_{1,n} \simeq (TL)^{d-1} \,, \qquad\qquad \text{for } T \gtrsim m_n \,,$$

$$N_n = \mathcal{Z}_{1,n} \simeq e^{-m_n/T} P_q\left(\frac{m_n}{T}\right)(TL)^{d-1} \,, \qquad \text{for } T \lesssim m_n \,, \tag{28}$$

where $P_q(x)$ represents a polynomial function on $x$ of degree $q \leq (d-1)/2$.

From here we see that species with masses $m_n \lesssim T$ have a typical occupation number that at leading order does not depend significantly on their mass, whereas for masses above $T$ there exists the expected exponential suppression. This effectively means that all species with masses up to the order of the temperature will contribute to the partition function, whereas the ones with masses above such temperature will be exponentially suppressed and effectively will not contribute (unless the degeneracy of such species can compensate the exponential suppression, as is the case for stringy states, which we consider separately below).

Thus, for a given temperature, $T$, such that the number of species with masses below such temperature is sufficiently large, we can compute the average number of weakly coupled one-particle states that are occupied, as given in eq. (24), by approximating the sum by an integral

$$\langle N_{\rm TOT}(T) \rangle = \int_0^\infty dm\, \rho(m) N(m)$$

$$= (TL)^{d-1} \int_0^T dm\, \rho(m) + (TL)^{d-1} \int_T^\infty dm\, e^{-m_n/T} P_q\left(\frac{m_n}{T}\right) \rho(m) \,. \tag{29}$$

Here we have expressed everything in terms of the masses, namely $N_n = N(m_n)$, and defined the density of (species) states simply as $\rho(m) = \rho(n)dn/dm_n$. The first integral counts the number of species below temperature $T$, which resembles our definition of $N_T$ in the previous section

$$N_T = \int_0^T dm\, \rho(m) \,. \tag{30}$$

For a spectrum with polynomial degeneracy, which we can parameterize as in (9), this reduces to

$$N_T = \left(\frac{T}{m_{\mathrm{t}}}\right)^p , \qquad (31)$$

which coincides with eq. (11). The integral between $T$ and $\Lambda_{\mathrm{sp}}$ gives the following result

$$\left(\frac{T}{m_{\mathrm{t}}}\right)^p \; p \; [\Gamma(q+p,1)] , \qquad (32)$$

where we have use the same parameterization for the mass spectrum and also the fact that the polynomial $P_q(m_n/T)$ has degree $q \leq d-2$. The factor multiplying $N_T$ gives an extra term proportional to $N_T (TL)^{d-1}$. All in all, for towers with finite $p$,[9] the parametric dependence of the average number of occupied one-particle states is given by

$$\langle N_{\mathrm{TOT}}(T)\rangle \sim N_T (TL)^{d-1} , \qquad (33)$$

which can be interpreted as the number of species below temperature $T$, captured by $N_T$, times the corresponding number of available momentum states for each of these species, which is of order $(TL)^{d-1}$. Since the average number of one-particle states that are populated effectively defines the average number of particles in the box, it is crucial to understand also the fluctuations in such number. In particular, using the definition of the fluctuation (A.10), we obtain for the system defined by the partition function (26) the following expression

$$\frac{\Delta N_{\mathrm{TOT}}}{\langle N_{\mathrm{TOT}}\rangle} \sim \frac{1}{\sqrt{\langle N_{\mathrm{TOT}}\rangle}} \sim \frac{1}{\sqrt{N_T (TL)^{d-1}}} . \qquad (34)$$

As expected, the distribution becomes arbitrarily narrow as $N_{\mathrm{TOT}}$ grows, which will always be the case for the configurations we study (either because $N_T$ becomes large or because $T \gg L$). In fact, this is also valid for the average number of one-particle states of each species, namely $\Delta N_n \sim \sqrt{N_n}$. Therefore, for large $N_{\mathrm{TOT}}$ we can simply use the average values up to order one factors that we are not important for our computations.

In such cases, given that $\mathcal{Z}_{1,n} = N_n \sim (TL)^{d-1}$ for all $n \lesssim N_T$, we can approximate the logarithm of the partition function (neglecting subleading additive contributions from $n > N_T$) (26) by

$$\log \mathcal{Z} \simeq N_n N_T \log(N_n) - N_T \log(N_n!) . \qquad (35)$$

We will be interested in the limits where $T \gg 1/L$ or $T \simeq 1/L$, and it turns out that in both cases we can rewrite the log of the partition function as

$$\log(\mathcal{Z}) \sim N_{\mathrm{TOT}} , \qquad (36)$$

where in the former case one can use Stirling's approximation for the denominator and in the later we use that $N_n \sim 1$ for $T \simeq L$.[10]

With this simple dependence of the partition function on the average number of one-particle states that are populated at temperature $T$, it is straightforward to compute the average energy and its corresponding relative fluctuation, which take the form

$$\langle E \rangle \sim N_{\mathrm{TOT}} T , \qquad \frac{\Delta E}{\langle E \rangle} \sim \frac{1}{\sqrt{N_{\mathrm{TOT}}}} , \qquad (37)$$

---

[9]The case $p \to \infty$, which can be though of an effective parameterization of the stringy regime, will be discussed separately below.

[10]Note that in the case $T \simeq L$ we are approximating $N_n \sim 1$ to indicate that there is no asymptotic dependence in any of the variables of the problem, but keeping in mind that it $N_n \neq 1$ in order to avoid the vanishing of the logarithm, which would not make sense from the point of view of e.g. entropy counting.

where we have used eqs. (A.11) and (A.12).

Finally, let us compute the entropy, which reads

$$S = \log(\mathcal{Z}) + T\,\frac{\partial \log(\mathcal{Z})}{\partial T} \sim N_{\text{TOT}}\,, \tag{38}$$

where in the last step we have used that $\log(\mathcal{Z}) \sim \log N_{\text{TOT}} \sim T^p$, as is the case for the spectrum in eq. (9).

Let us now comment on the different asymptotic behaviours of the entropy (i.e. $N_{\text{TOT}}$) and the energy for the different hierarchies of control parameters that we will consider throughout the rest of the paper

- **Thermodynamic limit:** $\Lambda_{\text{sp}} \geq T \gg 1/L$

  In this case, the entropy grows with the volume. All the species with masses of order $m_n \lesssim T$ effectively contribute to the entropy with a contribution proportional to the volume, i.e. $(TL)^{d-1}$. The energy is mainly dominated by the kinetic energy of the species with masses below $T$, and thus we obtain $E \sim N_{\text{TOT}}\, T$. These configurations are the ones that are constrained by the CEB and the gravitational collapse bound, as explained in section 2, and this imposes a stricter upper bound on the maximum allowed temperature, which cannot be arbitrarily close to $\Lambda_{\text{sp}}$ (c.f. eq. (13).

- **Frozen momentum limit:** $\Lambda_{\text{sp}} \gg T \simeq 1/L$

  In this limit, each of the species contributing to $N_T$ is effectively frozen in $d$-dimensions, since the number of available $d$-dimensional momentum states that fit in the box of size $L$ and remain below $T$ is of order one. Thus, $N_{\text{TOT}} \simeq N_T$ and the entropy comes mainly from the number of species with masses $m_n \lesssim T$. The average energy takes the form $E \sim N_T\, T$, which we can also rewrite as $E \sim \sum_{n=1}^{N_T} m_n$ using a parameterization for the mass spectrum as the one in eq. (9), together with the definition of $N_T$ given in (31). Note that this resembles the species energy introduced in [26], but with contributions from all species below $T$, instead of $\Lambda_{\text{sp}}$. These configurations do not collapse gravitationally nor exceed any holographic entropy bounds since $T < \Lambda_{\text{sp}}$, due to the fact that the volume contribution to the entropy and the energy is effectively frozen by setting the lowest possible temperature, namely $T \simeq 1/L$.

- **Species scale limit:** $\Lambda_{\text{sp}} \simeq T \simeq 1/L$

  Finally, we can consider the previous case, where the $d$-dimensional degrees of freedom are effectively frozen, at the same time as we increase the temperature (and consequently decrease the size of the box) until $\Lambda_{\text{sp}} \simeq T \simeq 1/L$. As argued in section 2, this is the limiting case in which the gravitational collapse bound and the CEB coincide, as can be seen by the fact that the entropy scales like $S \sim N_{\text{sp}} \sim (M_{\text{Pl},d}\,L)^{d-2}$, which recovers the area law for the entropy in the limit in which the system would collapse to the smallest possible black hole in the EFT, namely $L \simeq \Lambda_{\text{sp}}^{-1}$. The average energy of the system, which takes the form $E \sim N_{\text{sp}}\,\Lambda_{\text{sp}}$, can also be re-expressed as the species energy [26], $E \sim \sum_{n=1}^{N_{\text{sp}}} m_n$, which from this point if view it is nothing but the average energy of the system of species in the canonical ensemble, since all of them have average order-one occupation number.

**One-particle state domination**

We now consider the opposite regime, in which the partition function is dominated by configurations in which the total energy of the system is mainly accounted for by a single particle

with $E \simeq m_n$. In order to compute thermodynamic quantities we need the logarithm of $\mathcal{Z}$, which in this case can be approximated by

$$\mathcal{Z} \simeq \sum_{n=1}^{\infty} e^{-\frac{m_n}{T}} d_n \,, \tag{39}$$

where $d_n$ is the degeneracy at each level. The validity of our thermodynamic description should at least have a convergent partition function, otherwise we would have that our original degrees of freedom are not an appropriate description of the system. This motivates three different regimes in terms of the degeneracy of the levels. On the one hand, for subexponential degeneracies we have that $\mathcal{Z}_1$ converges for every temperature since the exponential suppression makes the contributions from heavy one-particle states subdominant. In this case, $\mathcal{Z} \gg \mathcal{Z}_1$ and (26) is the right approximation for the partition function, instead of (39). For completeness, let us mention that this is also the relevant regime in the presence of an exponential degeneracy whenever the temperature is well below the Hagedorn temperature, such that only the massless modes are active and the exponential degeneracy is effectively invisible. On the other hand, if the degeneracy is exponential, such that $d_n \simeq e^{m_n/T_{\mathrm{H}}}$,[11] the partition function takes the form

$$\mathcal{Z} \simeq \sum_{n=1}^{\infty} e^{-m_n\left(\frac{1}{T} - \frac{1}{T_{\mathrm{H}}}\right)} \,, \tag{40}$$

and only converges for $T < T_{\mathrm{H}}$, where $T_{\mathrm{H}}$ is the Hagedorn temperature [58], which acts as a cutoff for the effective description. At temperatures near $T_{\mathrm{H}}$, the total partition function, $\mathcal{Z}$, is dominated by configurations in which most of the energy is stored in the mass of a one-particle state (in the string picture this corresponds to the fact that these configurations are dominated by single, long strings) [60].

For such tower we can compute the average energy and its fluctuation using eqs.(A.11) and (A.12) to obtain

$$\langle E \rangle \sim \frac{T_{\mathrm{H}}}{T_{\mathrm{H}} - T} \, T \equiv N_T \, T \,, \qquad \Delta E \sim \langle E \rangle \,, \tag{41}$$

while the entropy is given by

$$S \sim N_T \,. \tag{42}$$

Here, we have used the (leading contribution to the) entropy to define $N_T$, since unlike in the polynomially degenerate case, it does not directly correspond to the number of species with masses below $T$. Instead, in this case, the majority of states (or species) that contribute to the partition function at temperature $T$ (near, but below $T_{\mathrm{H}}$) have masses above $T$, but they still contribute due to the large degeneracy. Thus, it is not possible to give an intuitive definition from comparing $T$ with $m_n$ directly. However, given that in the familiar cases the entropy gives the correct notion of particle (or species) number, we can use it as a definition in the current case. Crucially, we have not assumed any particular scaling between the (average) energy and the particle number, but we obtain the same as in the polynomially degenerate case, which is pivotal for the validity of our arguments. In fact, this can be traced back to the fact that from the two contributions to the canonical entropy, namely the one proportional to $E/T$ and the one proportional to $\log(\mathcal{Z})$, the former is parametrically dominant in the exponentially degenerate case, whereas the two of them are parametrically equal in the polynomially degenerate one, yielding the $S \sim E/T$ scaling.

---

[11]The degeneracy can, and in general does, include multiplicative subexponential contributions such as $d_n \simeq \left(\frac{m_n}{T_{\mathrm{H}}}\right)^{\alpha} e^{m_n/T_{\mathrm{H}}}$, with $\alpha \geq 0$. These may differ between the one-particle and the multi-particle density of states, but the exponential piece remains the same and is the one that sets the maximum temperature for the canonical partition function to converge, $T_{\mathrm{H}}$.

Furthermore, in contrast to a polynomial spectrum, we note that the relative energy fluctuations do not decrease as we increase $T$ (or equivalently, $N_T$). This shows that, even though it can be analyzed in the canonical ensemble, the microcanonical analysis is not recovered in the high-entropy limit [60], which we study separately in the next subsection (see also [27, 29]).

Finally, let us comment on the limit when $T \to \Lambda_{\mathrm{sp}}$. Our strategy is to equate $N_T$ to $N_{\mathrm{sp}}$, which we define as $1/g_{s,d}^2$ for the case of a string tower, and solve for $\Lambda_{\mathrm{sp}}$, namely

$$\lim_{T \to \Lambda_{\mathrm{sp}}} N_T = N_{\mathrm{sp}} = \left( \frac{M_{\mathrm{Pl},d}}{\Lambda_{\mathrm{sp}}} \right)^{d-2} = \frac{1}{g_{s,d}^2}. \tag{43}$$

Near $T_{\mathrm{H}}$, or equivalently for a large $N_T$, the leading term for the species scale is given by

$$\Lambda_{\mathrm{sp}} \sim T_{\mathrm{H}}, \tag{44}$$

where the subleading additive contribution vanishes in the $g_{s,d} \to 0$ limit, and it is compared with the general form of the species scale in section 3.3 (c.f. eq. (105)). This matches the expected behavior for a string tower, where in the weak coupling limit $\Lambda_{\mathrm{sp}} \sim T_{\mathrm{H}} \simeq m_{\mathrm{str}}$, and it is clear that the species scale is capturing the maximum possible temperature that can be described by the EFT, given by the maximum temperature for which the canonical partition function is convergent.

**Super-exponential degeneracy and divergence of the canonical partition function**

Finally, we note that for $d_n$ growing parametrically faster than an exponential (i.e. for $d_n e^{-m_n/T_0} \to \infty$ as $n \to \infty$), the partition function does not converge for any $T > 0$. This allows us to rule out towers with super-exponential degeneracy, since systems with such high degeneracies lead to a divergent canonical partition function and a breakdown of our original description. Similarly, the average energy diverges, suggesting a mismatch between the canonical and microcanonical descriptions.

In this regard, let us mention that if one decided to introduce by hand a UV regulator in the masses that contribute to the canonical partition function by cutting off the tower at a maximum $N$, which we could then parameterize as

$$\mathcal{Z} \simeq \sum_{n=1}^{N} e^{-\frac{m_n}{T} + \left( \frac{m_n}{T_0} \right)^{\alpha}}, \tag{45}$$

for some $\alpha > 1$ (such that the spectrum is superexponential). The introduction of a cutoff then implies finite average energy, and a matching microcanonical description. From the partition function it is plain to see that it would be dominated by the last term only, this due to the fact that the degeneracy grows so fast that we have $d_n \ll d_{n+1}$, and thus

$$\mathcal{Z} \simeq e^{-\frac{m_N}{T} + \left( \frac{m_N}{T_0} \right)^{\alpha}} = N_T \, e^{-\frac{m_N}{T}}. \tag{46}$$

This spectrum then corresponds to $N_T$ degenerate states with mass $m_N$, and as we discuss in detail in section 3.3, it can be thought of as an effective parameterization of the exponentially degenerate case if $\Lambda_{\mathrm{sp}} \simeq m_N$.

## 3.2 Microcanonical ensemble

We have seen that for towers of states with polynomial degeneracy the canonical ensemble yields relative fluctuations on the energy that asymptote to zero as the entropy (or, equivalently, the temperature and $N_{\mathrm{TOT}}$) is increased, effectively recovering the microcanonical ensemble

in that limit. For exponentially degenerate towers, however, it is known that as we approach the limiting Hagedorn temperature the relative fluctuations do not decrease (c.f. eq. (41)) and thus the canonical ensemble does not reduce to the microcanonical one [60].

Independently of the size of the fluctuations, we have obtained that in the relevant limits the thermodynamic quantities take the following form

$$E \simeq N_T T, \qquad S \simeq N_T, \tag{47}$$

which converge to the constitutive relations of species thermodynamics in the limit $T \to \Lambda_{\rm sp}$.

With this in mind, it is illustrative to study the problem in the strict limit $\Lambda_{\rm sp} = T = 1/L$ in the microcanonical ensemble, in which we expect to recover the same constitutive relations for the energy and entropy [27]. To do so, we consider a tower of particles with masses $m_n \leq T = \Lambda_{\rm sp} = 1/L$ and follow [27] to define the number[12]

$$\mathcal{M} = E/m_{\rm t}. \tag{48}$$

Where $E$ is the total energy, and $m_{\rm t}$ is the mass of the lightest particle in the tower. The problem is now to find all possible combinations with total energy $E = \mathcal{M}m_{\rm t}$, from summing masses $m_n = f(n)m_{\rm t}$ (with $f(n)$ a function of $n$, c.f. (9)), with degeneracy $d_n$ and below the cutoff scale $\Lambda_{\rm sp}$. Note that this approach is slightly different from the canonical one in the sense that we are now considering only contributions from species with masses below the given $T = \Lambda_{\rm sp}$, whereas in the canonical ensemble the particles with masses above $T$ could still be the dominant contribution for sufficiently high degeneracies (i.e. exponential or higher).[13] That is, we consider only $n \leq N_{\rm max}$, with $N_{\rm max}$ the maximum level defined via

$$\Lambda_{\rm sp} = f(N_{\rm max})m_{\rm t}, \qquad N_{\rm sp} = \sum_{n=1}^{N_{\rm max}} d_n, \tag{49}$$

where we are also giving its relation to the number of species below $\Lambda_{\rm sp}$. The number of such combinations, that we denote $\Omega(\mathcal{M}, N_{\rm max})$, is known to have generating function [61]

$$Z(q) = \sum_M \Omega(\mathcal{M}, N_{\rm max})q^{\mathcal{M}} = \prod_{n \leq N_{\rm max}} \frac{1}{\left(1 - q^{f(n)}\right)^{d_n}}. \tag{50}$$

The desired combination can then be obtained by contour integration as

$$\Omega(\mathcal{M}, N_{\rm max}) = \oint \frac{dq}{q^{\mathcal{M}+1}} Z(q). \tag{51}$$

For many species below the cutoff it is reasonable to assume $\mathcal{M} \gg N_{\rm max}$, such that the poles at $q = 0$ do not contribute to the integral. Following the procedure in [27] one finds

$$S = N_{\rm sp} + N_{\rm sp} \log\left(\frac{\mathcal{M}}{N_{\rm sp}}\right) - \sum_n^{N_{\rm max}} d_n \log f(n) + \mathcal{O}(\log \mathcal{M}). \tag{52}$$

The temperature is then, for a general tower at $T \simeq \Lambda_{\rm sp}$

$$T = \left(\frac{\partial S}{\partial E}\right)^{-1} = \frac{E}{N_T} = \frac{E}{N_{\rm sp}} = \Lambda_{\rm sp}. \tag{53}$$

---

[12]Strictly speaking, $\mathcal{M}$ should be an integer, but since for a large number of species we have that $E \gg m_{\rm t}$ we can neglect the subtleties associated to this.

[13]This is related to the logarithmic ambiguity in the definition of the species scale for stringy towers that arises when counting species as opposed to the entropy of the minimal black hole of size $R_{\rm BH} \sim \ell_{\rm str}$, as discussed in e.g. [16–19, 27, 28].

We also define the species chemical potential as

$$\mu \equiv -\Lambda_{\text{sp}} \frac{\partial S}{\partial N_{\text{sp}}} , \qquad (54)$$

such that $\mu = 0$ implies maximum entropy and an equilibrium configuration. We can estimate the chemical potential for a general tower by first approximating the third term in (52) by an integral and using

$$\frac{\partial}{\partial N_{\text{sp}}} \int_1^{N_{\text{max}}} dn \, d_n \log(f(n)) \simeq \frac{1}{d_{N_{\text{max}}}} \frac{\partial}{\partial N_{\text{max}}} \int_1^{N_{\text{max}}} dn \, d_n \log[f(n)] = \log f(N_{\text{max}}). \qquad (55)$$

The chemical potential then takes the form

$$\mu \simeq -\Lambda_{\text{sp}} \log \frac{\mathcal{M}}{N_{\text{sp}} f_{N_{\text{max}}}} , \qquad (56)$$

and the entropy will be maximized for configurations satisfying

$$\mathcal{M} \simeq N_{\text{sp}} f(N_{\text{max}}), \qquad S \simeq N_{\text{sp}} + \sum_{n=1}^{N_{\text{max}}} d_n \log \frac{f(N_{\text{max}})}{f(n)} . \qquad (57)$$

In general, these configurations of maximal entropy are then bounded as

$$N_{\text{sp}} \leq S \leq N_{\text{sp}} \left( 1 + \log\left( \frac{\Lambda_{\text{sp}}}{m_{\text{t}}} \right) \right), \qquad (58)$$

where we have used $T/m_{\text{t}} = f_N/f_1$ and assumed that $f_{n+1} \geq f_n$. We can note however, that for sufficiently high degeneracy the sum will be dominated by the $n = N_{\text{max}}$ limit, such that the contribution of the sum vanishes and the entropy is

$$S \simeq N_{\text{sp}} . \qquad (59)$$

This is the case for exponential (and higher) degeneracies, up to small (i.e. logarithmic) corrections, where $m_{\text{t}} \sim \Lambda_{\text{sp}}$ and the upper bound is saturated. In contrast, towers with polynomial degeneracy can never saturate this bound.

## 3.3 Appropriate towers and species entropies

The goal of this section is to systematically study different kinds of towers in the canonical ensemble and determine which of them can give rise to the correct scaling of the entropy and energy in the limit $T \to \Lambda_{\text{sp}}$ to reproduce the ones of the corresponding minimal black hole (we elaborate more on this picture, relating this behaviour to the correspondence between black holes and the thermal ensemble, in section 5). To that end, and inspired by the two cases studied in section 3.1 (which correspond to the only two types of consistent towers in Quantum Gravity according to the Emergent String Conjecture [10], namely KK-like and weakly coupled string towers) we analyze more general spectra and are able to rule out the ones that do not correspond to the ones previously analyzed.

Our starting point is to consider a tower of states with masses $m_n$, such that when the states are thermalized at the maximum allowed temperature, $T \simeq \Lambda_{\text{sp}} = M_{\text{Pl},d}/N_{\text{sp}}^{\frac{1}{d-2}}$, precisely $N_{\text{sp}}$ of them effectively contribute to the canonical partition function. Motivated by the results of the previous sections, we start from the following form for the energy of such configurations

$$E = N_T \, T , \qquad (60)$$

with $N_T$ is the number of states that contribute at temperature $T$ (e.g. the number of states with masses $m_n \lesssim T$ in the case of a tower with polynomial degeneracy). In general, recall that we can relate the energy, the entropy and the partition function via

$$E = T^2 \frac{\partial}{\partial T} \log(\mathcal{Z}), \qquad S = \frac{\partial}{\partial T}(T \log(\mathcal{Z})). \tag{61}$$

So that for the aforementioned form of the energy, the entropy and partition function satisfy

$$\partial_T \log \mathcal{Z} = \frac{N_T}{T}, \tag{62}$$

$$S = N_T + \log(\mathcal{Z}). \tag{63}$$

Then, a species distribution with energy (60) is *appropriate* if it fulfills the following two conditions

i. The number of kinematically available species does not decrease as we increase the temperature, $\partial N_T / \partial T \geq 0$.

ii. In the limit in which the momentum states available per species are of order one (i.e. $T \simeq 1/L$ for particles in a box), the entropy is given by $S \simeq N_T$, with $N_T \gg 1$ the number of species that are active at such temperature. Then, in the limit $T \to \Lambda_{\rm sp}$, the minimal black hole entropy is recovered.

In the rest of this section, we start by first considering what kinds of towers can mimic different parameterizations of the entropy that yield the aforementioned scaling. We then proceed to the systematic study of different kinds of towers following the logic outlined above, and concluding whether they are *appropriate* towers that can give rise to the right scaling of entropy and energy (particularly in the limit $T \to \Lambda_{\rm sp}$) or not. We also revisit the cases of the KK-like and string-like towers following this logic to make the discussion more complete and easy to follow, even though we know from the get go that they fulfill the right properties as explained in section 3.1. We perform this analysis in the canonical ensemble, by considering thermal configurations of species at temperature $T$, but equivalent computations in the microcanonical ensemble are found in Appendix B.

**Corrections to the entropy and structure of the towers**

We begin by exploring what kind of modifications in the structure of the tower (i.e. its degeneracies, $d_n$) can account for different multiplicative or additive corrections for the entropy. First, we consider the following form for the entropy and the partition function

$$S = cN_T, \qquad \frac{\log(\mathcal{Z})}{N_T} = c - 1, \tag{64}$$

with $c$ an arbitrary, order-one constant. Note, in particular, that this kind of multiplicative correction does not change the parametric dependence of the entropy that we argued to be the *appropriate* one. To be precise, since we have not kept track of $\mathcal{O}(1)$ factors so far, a multiplicative correction of this type might seem irrelevant. However, the point that we want to emphasize is that in this context it encapsulates the fact that both contributions to the entropy in (63) are parametrically the same, i.e. $N_T \sim \log(\mathcal{Z})$, and this implies that $N_T$ grows polynomially in $T$, as we show in the following. This form of the entropy corresponds to a number of states

$$N_T = \left(\frac{T}{T_0}\right)^{\frac{1}{c-1}}. \tag{65}$$

Then, for $c > 1$ we have that the number of states entering the theory increase with the temperature, and both conditions are satisfied. Here and in the rest of this paper, we consider a tower of states with mass scaling

$$m_n = n^{1/p} m_{\mathrm{t}},\qquad(66)$$

and we can then easily compute the degeneracy

$$d_n = n^{\frac{p}{c-1}-1}.\qquad(67)$$

In order to reproduce $S \simeq N_T$, the contribution coming from $\log(\mathcal{Z})$ has to be at most, of the same order as $N_T$. Thus, we conclude that polynomial towers lead to $\log(\mathcal{Z}) \sim N_T$.

Let us now turn to the structure of towers that produce a subdominant contribution coming from $\log(\mathcal{Z})$, namely $\log(\mathcal{Z}) \ll N_T$ for large $N_T$, such that the leading form of the entropy is still the *appropriate* one and we recover the right result for large number of species. We first consider additive, subleading, polynomial corrections to the entropy, which we parameterize as

$$S = N_T\left(1 + \frac{a}{N_T^k}\right),\qquad \log(\mathcal{Z}) = aN_T^{1-k},\qquad(68)$$

with $k > 0$ and $a = \mathcal{O}(1)$. Here we have assumed $N_T$ continuous and differentiable in the range $T > T_0$. The number of states takes the following form

$$N_T \simeq \frac{1}{\log(T/T_0)^{1/k}},\qquad(69)$$

and since $N_T$ decreases with the temperature, it is clear the first condition is not satisfied. This suggests that such polynomially suppressed corrections do not come from *appropriate* towers of states. In fact, one can check that any correction with polynomial or super-polynomial suppression will violate this first condition, so we do not consider them any further.

We can then consider corrections to the entropy with sub-polynomial suppression, namely

$$S = N_T\left(1 + \frac{a}{\log(N_T)}\right),\qquad \log\mathcal{Z} = a\,\frac{N_T}{\log(N_T)}.\qquad(70)$$

The corresponding number of states and degeneracy are

$$N_T \simeq e^{T^{1/a}},\qquad d_n \simeq e^{n^{1/(ap)}},\qquad(71)$$

such that both conditions are satisfied. In general, corrections to the entropy of the form

$$S = N_T\left(1 + \frac{a}{\log^{[k]}(N_T)}\right),\qquad \log(\mathcal{Z}) = a\,\frac{N_T}{\log^{[k]}(N_T)},\qquad(72)$$

with $\log^{[k]}(x) \equiv \log\log\log\ldots_{(k\text{ times})}\ldots\log(x)$, also lead to *appropriate* towers with

$$N_T \simeq \exp\exp\exp\ldots_{(k\text{ times})}\ldots\exp\left(T^{1/p}\right),\qquad d_n \simeq \exp\exp\exp\ldots_{(k\text{ times})}\ldots\exp\left(n^{1/(ap)}\right).\qquad(73)$$

Note that polynomial and exponential degeneracies are special cases of (73), with $k = 0$ and $k = 1$, respectively. Furthermore, as mentioned in section 3.1, super-exponential towers (i.e. $k > 1$) have a divergent canonical partition function for any non-vanishing temperature, so they do not represent can also be ruled out as the fundamental, weakly coupled degrees of freedom in the theory, so we also do not consider these as *appropriate*.

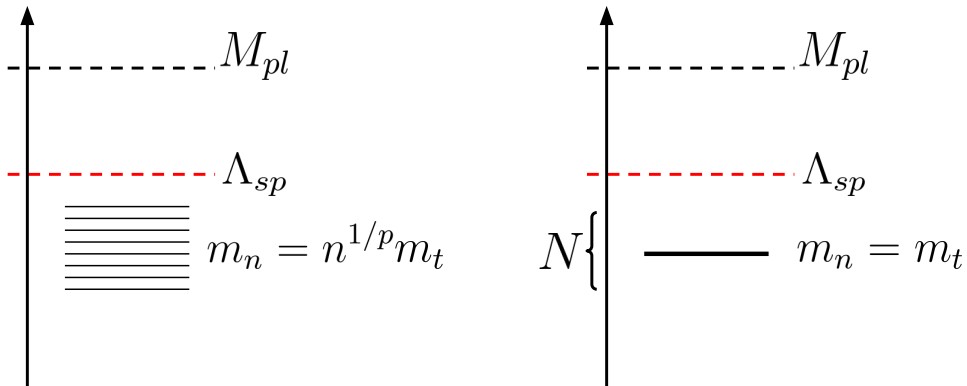

Figure 1: Left (Right):Tower of states for polynomial (constant) degeneracy.

### 3.3.1 Tower with polynomial degeneracy

We now turn to the systematic study of different spectra for the towers. Let us begin by considering a tower that we know arises in Quantum Gravity setups, namely one with the following mass spectrum and density of states:[14]

$$m_n = n^{1/p} m_{\mathrm{t}}, \qquad d_n = 1. \tag{74}$$

Such a tower corresponds to Kaluza-Klein compactification on an isotropic $p$ dimensional manifold, e.g. a torus $\mathcal{T}^p$, and at temperature $T$ there are

$$N_T = \left(\frac{T}{m_{\mathrm{t}}}\right)^p, \tag{75}$$

available species. The structure of such tower is shown in the left side of Fig. 1.

The energy is then given by

$$E = N_T T = \frac{T^{p+1}}{m_{\mathrm{t}}^p}, \tag{76}$$

and using (61) we can compute the corresponding partition function and entropy

$$\log(\mathcal{Z}) = \left(\frac{T}{m_{\mathrm{t}}}\right)^p \frac{1}{p} = \frac{N_T}{p}, \qquad S = \frac{T^p}{m_{\mathrm{t}}^p} \frac{p+1}{p} = N_T \frac{p+1}{p}. \tag{77}$$

Using eq. (65) we can identify $c = \dfrac{p+1}{p}$, and we obtain that the tower fulfills the requirements presented above and can be considered as an *appropriate* tower, as expected for a KK-like tower.

In the high-temperature limit $T \simeq \Lambda_{\mathrm{sp}}$ we have

$$S = N_{\mathrm{sp}} \frac{p+1}{p} = \left(\frac{\Lambda_{\mathrm{sp}}}{m_{\mathrm{t}}}\right)^p \frac{p+1}{p}. \tag{78}$$

Also, recall that the string limit is recovered in the limit $p \to \infty$, such that

$$m_{\mathrm{t}} = \Lambda_{\mathrm{sp}}, \quad S = N_{\mathrm{sp}}. \tag{79}$$

In this limit one recovers the free string entropy, as the UV cutoff approaches the mass of the tower $\Lambda_{\mathrm{sp}} = m_{\mathrm{t}}$,

$$S = \frac{E}{m_{\mathrm{t}}}, \qquad \frac{\log \mathcal{Z}}{N_T} \simeq 0, \tag{80}$$

and the Hagedorn temperature can be defined as $T_{\mathrm{H}} \simeq m_{\mathrm{t}}$.

---

[14]This is equivalent to a tower with $m_n = n m_{\mathrm{t}}, d_n = n^{p-1}$.

**Negative $p$ and black holes**

As a side comment, let us note that the family of solutions in eq. (75) can also effectively parameterize the entropy and energy associated to $d$-dimensional black holes if one considers negative values for $p$ of the form

$$p = -(d-2), \tag{81}$$

for $d > 3$. From here we obtain the following form for the entropy (and partition function)

$$S = \frac{d-3}{d-2} T^{-(d-2)} \propto E^{\frac{d-2}{d-3}}, \qquad \frac{\log \mathcal{Z}}{N_T} = -\frac{1}{d-2}, \tag{82}$$

which are the expected relations for a black hole with ADM mass $M = E$ in $d$ dimensions. At this point, we would like to highlight that this is simply an observation whose potential physical meaning is beyond the scope of this work. In summary, we can classify the three different regimes in terms of the exponent $p$ as

$$\begin{aligned} 1 \leq p < \infty\,, \qquad & \text{KK tower,} \\ |p| \to \infty\,, \qquad & \text{string tower/highly excited string,} \\ -\infty < p \leq -1\,, \qquad & \text{black hole microstates.} \end{aligned} \tag{83}$$

In accordance with our prescription, the black hole microstates cannot be considered as a fundamental, weakly-coupled, tower of states, as the microstate count decreases with the temperature. Furthermore, we can discard the case $-1 < p < 0$, since it implies $S < 0$. The case $0 < p < 1$, effectively parameterizes sub-polynomial, i.e. decreasing degeneracy, since for a linear mass spectrum $m_n = m_t n$ one has $d_n = n^{p-1}$, such that $d_{n+1} < d_n$. Similarly, the case of $p = 0$ as parameterized here involves states siting at a fixed mass $m_n = m_t$, so we consider these cases separately in the following.

### 3.3.2 Tower with sub-polynomial and constant degeneracy

We consider now a "tower" with the following spectrum:

$$m_n = n\, m_t, \qquad d_n = \delta_{n,1} N\,. \tag{84}$$

This corresponds to $N$ states accumulating at a mass $m_t$. One could think of this, for example, as a situation with $\sim N$ non interacting copies of the Standard Model [13].[15] This is shown in the right side of Fig. 1. For $T \geq m_t$ the energy is given by

$$E = N T\,, \tag{85}$$

and the partition function and the entropy then take the form

$$\log(\mathcal{Z}) = N \log\left(\frac{T}{m_t}\right), \qquad S = N \left\{1 + \log\left(\frac{T}{m_t}\right)\right\}. \tag{86}$$

For $T \simeq \Lambda_{sp}$ we find

$$S = N \left\{1 + \log\left(\frac{\Lambda_{sp}}{m_t}\right)\right\}. \tag{87}$$

As such, states accumulating around a single mass do not give the correct dependence between entropy and number of species, as they present large corrections for $T \gg m_t$. Remarkably, however, the correct entropy can be recovered for $m_t \simeq \Lambda_{sp}$, corresponding to the string limit.[16]

---

[15]Additionally, as mentioned in section 3.1, one can also interpret this as towers with a super-exponential degeneracy in the spectrum and maximum level given by $N$, upon identification of $m_N \simeq \Lambda_{sp}$, such that $d_n \ll d_{n+1}$ and the entropy is dominated by the species occupying the last level.

[16]Similarly, this suggests that a super-exponential tower accompanied by a UV cutoff effectively parameterizes a string-like tower.

Away from this limit, there is an obstruction to increasing the number of species above

$$N_{\mathrm{crit}} = \left(\frac{M_{\mathrm{Pl}}}{m_{\mathrm{t}}}\right)^{d-2}, \tag{88}$$

since for $N > N_{\mathrm{crit}}$ the EFT description including any state of the tower would be invalid, as the lightest one would already be above the UV cutoff.

The case of sub-polynomial towers is deeply related to fixed mass spectra. These sort of towers have decreasing degeneracy $d_n > d_{n+1}$. It is important to recall here that $d_n$ are positive integers, since this plays a key role when $d_n$ is a monotonically decreasing function of $n$, as opposed to the case in which it is monotonically increasing and can be approximated by a real function of $n$ (as we have done repeatedly in the cases studied above). Therefore, we can define the maximum level $N_{\mathrm{max}}$ such that $d_{N_{\mathrm{max}}+1} = 0$. We assume a large initial degeneracy $d_1 \gg 1$, since the case $d_1 \simeq \mathcal{O}(1)$, $d_2 = 0$ is trivially a fixed mass tower. The simplest non-trivial case to understand is when the degeneracy decreases very fast, i.e. $d_n \gg d_{n+1}$, and $m_t \leq \Lambda_{\mathrm{sp}}$. In such situation, the entropy will be dominated by the first level and behave like the fixed mass spectrum analyzed above. In fact, as long as the spectrum is such that $m_{N_{\mathrm{max}}} < \Lambda_{\mathrm{sp}}$, the behavior of such sub-polynomial tower can be effectively parameterized by that of the fixed mass spectrum presented in (84) with the following identifications

$$N = \sum_{n=1}^{N_{\mathrm{max}}} d_n, \quad m_t = \frac{1}{N} \sum_{n=1}^{N_{\mathrm{max}}} m_n d_n. \tag{89}$$

As we have shown around eq. (87), since $\Lambda_{\mathrm{sp}} > m_{N_{\mathrm{max}}} > m_{\mathrm{t}}$ this could not give rise to an *appropriate* tower unless we have a very compressed spectrum that effectively captures a constant degeneracy tower with $\Lambda_{\mathrm{sp}} \simeq m_{N_{\mathrm{max}}} \simeq m_{\mathrm{t}}$.

On the other hand, if the degeneracy decreases slowly enough, such that that $m_{N_{\mathrm{max}}} > \Lambda_{sp}$, one can expand the number of states at temperature $T < \Lambda_{\mathrm{sp}}$ as

$$N_T = \sum_{n=1}^{T/m_t} d_n \simeq \frac{T}{m_t} d_n \bigg|_{n=\frac{T}{m_t}} + \frac{T^2}{m_t^2} \frac{\partial d_n}{\partial n} \bigg|_{n=\frac{T}{m_t}} \cdots \tag{90}$$

The spectrum will then behave like a tower with constant degeneracy ($p = 1$), up to corrections that remain small as long as

$$\left|\frac{\partial d_n}{\partial n}\right| \ll d_n \left(\frac{T}{m_t}\right)^{-1} \bigg|_{n=T/m_t}. \tag{91}$$

In this sense, sub-polynomial spectra can be consistent in two ways. First, if the degeneracy decreases fast enough, such that $\Lambda_{\mathrm{sp}} \simeq m_{N_{\mathrm{max}}} \simeq m_{\mathrm{t}}$, which is effectively indistinguishable from a string-like tower ($p \to \infty$). Note that the tower with $N$ states accumulating at a single mass scale is a limiting case of this, for which $m_{N_{\mathrm{max}}} = m_{\mathrm{t}}$ (with $N_{\mathrm{max}} = 1$) by definition and thus they are only consistent if additionally $m_{\mathrm{t}} \simeq \Lambda_{\mathrm{sp}}$. Second, if their degeneracy decreases slowly enough to effectively resemble a spectrum with constant degeneracy for the masses that lie below $\Lambda_{\mathrm{sp}}$, which is effectively indistinguishable from a KK-like degeneracy with $p = 1$.

Finally, let us note that if we want to embed the previous discussion in the bigger picture of towers of states with masses depending on the moduli, as is the case in Quantum Gravity EFTs, the fact that the number of species must increase with $T$ in an *appropriate* way is deeply related with it having an *appropriate* dependence on the moduli whose vev controls the mass scales of the towers. Otherwise, we could not have the right relation between the entropy and the mass as we move towards infinite distance limits in moduli space. We will elaborate further on this idea and its connection to the black hole - tower correspondence in section 5.

### 3.3.3 Tower with exponential degeneracy

Lastly, let us consider a tower characterized by the following spectrum:

$$m_n = n^{1/\beta} m_t, \qquad d_n = e^{\lambda n^{1/\alpha}}, \tag{92}$$

with $0 < \alpha < \beta < 1$ and $\lambda > 0$, such that the canonical partition function converges for every temperature. As mentioned in section 3.1, the case $\alpha = \beta$ has to be treated separately, while $\alpha > \beta$ implies a divergent partition function for every $T > 0$ so we discard it from the get go. The level of the tower for which the mass equals the temperature is

$$N_{\max} = \left(\frac{T}{m_t}\right)^\beta, \tag{93}$$

and the number of species that contribute at such temperature (i.e. the ones with $n \leq N_{\max}$ can be computed from the sum $N_T = \sum_{n=1}^{N_{\max}} d_n$, which can be approximated by an integral for large $N_{\max}$ and yields

$$N_T \simeq e^{\lambda \left(\frac{T}{m_t}\right)^{\beta/\alpha}} \left\{ \frac{\alpha}{\lambda} \left(\frac{T}{m_t}\right)^{\beta \frac{\alpha-1}{\alpha}} + \frac{1}{2} \lambda^{2-\alpha} \right\}. \tag{94}$$

The energy can then be expressed as

$$E = N_T\, T \simeq m_t N_T \left(\frac{\log N_T}{\lambda}\right)^{\alpha/\beta}, \tag{95}$$

and the entropy and partition function then take the form

$$\log(\mathcal{Z}) \simeq \frac{\frac{\alpha}{\beta} N_T}{\log(N_T)} \ll N_T, \quad S \simeq N_T \left(1 + \frac{\alpha/\beta}{\log(N_T)}\right) \simeq \frac{E}{m_t \left(\log N_T^{1/\lambda}\right)^{\alpha/\beta}} \left(1 + \frac{\alpha/\beta}{\log N_T}\right). \tag{96}$$

In the large $N_T$ limit the entropy is then proportional to the number of species at temperature $T$. We note this is nothing but eq. (71), with $a = \alpha/\beta$, corresponding to an *appropriate* tower.

In the case $\alpha = \beta$, which includes a tower of string excitations when also $\alpha = 2$, we saw in section 3.1 that there is a competition of terms in the partition function

$$\mathcal{Z} = \sum_{n=1}^{\infty} e^{-m_{\mathrm{str}} n^{1/\alpha} \left(\frac{1}{T} - \frac{1}{T_H}\right)}, \tag{97}$$

where $T_H = \lambda\, m_{\mathrm{str}}$, with $\lambda$ some $\mathcal{O}(1)$ number that depends on the particular string theory under consideration. Convergence then requires $T < T_H$, and as a consequence there will be no light states in the tower as we had previously defined them. This occurs since the Hagedorn temperature is typically close to the mass of the tower $m_{\mathrm{str}} \simeq T_H \gtrsim T$. Performing the sum (by approximating it by an integral) yields

$$\mathcal{Z} \simeq \left(\frac{m_{\mathrm{str}}}{T} - \frac{m_{\mathrm{str}}}{T_H}\right)^{-\alpha}. \tag{98}$$

This reproduces the results of [32,60], where the partition function for the single highly energetic string is computed from the thermal scalar path integral of strings in $d$-dimensions. Our results, from simple thermodynamic considerations are only strictly valid in the limit in which momentum is frozen in the non compact directions, namely $d = 0$, but they match at leading order for any number of dimensions. The energy is then

$$E = \frac{\alpha\, T\, T_H}{T_H - T}, \tag{99}$$

while the free energy and the entropy are given by

$$\log(\mathcal{Z}) = \alpha \log\left(\frac{T\,T_{\mathrm{H}}}{m_{\mathrm{str}}\,(T_{\mathrm{H}} - T)}\right), \qquad S = \frac{\alpha T_{\mathrm{H}}}{T_{\mathrm{H}} - T} + \alpha \log\left(\frac{T\,T_{\mathrm{H}}}{m_{\mathrm{str}}\,(T_{\mathrm{H}} - T)}\right). \tag{100}$$

In the high temperature limit, $T_{\mathrm{H}} - T \ll T_{\mathrm{H}}$, we recover the expected relations between thermodynamic quantities for highly degenerate spectra

$$\frac{\log(\mathcal{Z})}{S} \to 0, \qquad \frac{E}{T} \to S. \tag{101}$$

In previous cases we had a direct interpretation of $N_T$ as the number of effectively light species, such that the exponential factor present in Boltzmann statistics was of $\mathcal{O}(1)$ for masses up to $m \lesssim T$ and exponentially suppressed above. However, as remarked in section 3.1, for a spectrum with an exponential degeneracy this interpretation is no longer valid. Given this, and eq. (101), we are motivated to define $N_T$ as the entropy for $T_{\mathrm{H}} - T \ll T_{\mathrm{H}}$, so that

$$N_T = \frac{1}{T_{\mathrm{H}} - T} \lim_{T \to T_{\mathrm{H}}} \{S\,(T_{\mathrm{H}} - T)\} = \frac{\alpha\,T_{\mathrm{H}}}{T_{\mathrm{H}} - T}. \tag{102}$$

$N_T$ and S then match near $T = T_{\mathrm{H}}$, and $N_T$ must be understood as the effective number of degrees of freedom that account for the entropy of the system. In terms of $N_T$ the energy and entropy take the expected form

$$E = N_T\,T, \qquad S = N_T\left(1 + \frac{\log(N_T)}{N_T}\right). \tag{103}$$

We note this can in principle be identified with eq. (69) with $k = 1$, but in the opposite regime of validity, for $T < T_0$, since $|\log T/T_0| \simeq 1 - T/T_0$. This is in accordance to out discussion on *appropriate* towers since near the singularity $N_T$ grows much faster than polynomially.

We can also obtain the species scale by solving

$$\lim_{T \to \Lambda_{\mathrm{sp}}} N_T = N_{\mathrm{sp}} = \left(\frac{M_{\mathrm{Pl},d}}{\Lambda_{\mathrm{sp}}}\right)^{d-2}, \tag{104}$$

near $T_{\mathrm{H}}$ and small string coupling, or equivalently for a large number of species. This yields

$$\Lambda_{\mathrm{sp}} \simeq T_{\mathrm{H}}\left(1 - \alpha\left(\frac{T_{\mathrm{H}}}{M_{\mathrm{Pl},d}}\right)^{d-2}\right). \tag{105}$$

Then, for a large but finite number of species we require $\Lambda_{\mathrm{sp}} < T_{\mathrm{H}} \ll M_{\mathrm{Pl},d}$, so that $\Lambda_{\mathrm{sp}} \simeq T_{\mathrm{H}}$.

For the known string case, $\alpha = 2$, we can use the definition of the string coupling, $g_s$, to write this as

$$\Lambda_{\mathrm{sp}} = T_{\mathrm{H}}\left(1 - 2\lambda^{d-2}g_s^2\right). \tag{106}$$

For Type II and Heterotic string theory the value of $\lambda$ has been explicitly computed as [62]

$$\lambda_{\mathrm{Type\ II}} = \frac{1}{\sqrt{2}}, \qquad \lambda_{\mathrm{Het}} = \left(1 + \frac{1}{\sqrt{2}}\right)^{-1}, \tag{107}$$

so that in e.g. $d = 4$ we have $2\lambda^2 = \mathcal{O}(1)$. This expression for the species scale is strictly smaller than the Hagedorn temperature, as the equality between the two only holds in the strictly weakly coupled limit in which $g_s^2 = 1/N_{\mathrm{sp}} = (\Lambda_{\mathrm{sp}}/M_{\mathrm{Pl},d})^{d-2} = 0$. This is in agreement with the discussion of [57], where it was argued that for a finite coupling gravitational effects cannot be neglected at high temperatures, and the system necessarily undergoes a phase transition before $T = T_{\mathrm{H}}$. Finally we would like to comment that the $g_s^2$ correction to $\Lambda_{\mathrm{sp}}$ also agrees with the leading order correction obtained from the Type II gravitational EFT [23, 24, 28].

All in all, by considering systems of species at equilibrium at temperature $T \rightarrow \Lambda_{\text{sp}}$, we have been able to rule out towers with different spectra. First, all towers with super-exponential degeneracies, are ruled out from the perspective of yielding a divergent canonical partition function for any finite $T$ (see also [27] for an alternative analysis of similar towers in the microcanonical ensemble). Furthermore, we can also rule out towers of $N$ species with fixed mass, unless this mass is precisely of order $\Lambda_{\text{sp}}$, which effectively resembles a tower of string excitations. Finally, we can also rule out towers with sub-polynomial spectra, since their degeneracy is decreasing they will have a finite spectrum, and they will be equivalent to fixed mass spectra. Then they can similarly be considered an effective parameterization of a string tower when their mass scale is parametrically close to $\Lambda_{\text{sp}}$. Consequently, only towers with polynomial or exponential degeneracy, which resemble KK-like and string-like towers, respectively, seem to give rise to the right form of the energy and entropy to match those of black holes as $T \rightarrow \Lambda_{\text{sp}}$, in agreement with the Emergent String Conjecture [10], and recent bottom-up arguments [27, 29].

## 3.4 The laws of species thermodynamics

The laws of species thermodynamics were first introduced in [26] in order to describe the statistics of towers at the high energy limit $T = \Lambda_{\text{sp}}$, in complete analogy to the usual laws of thermodynamics upon the proper identification of $S$, $T$ and $E$. Here we revisit their formulation, restating them in a form, which is more natural from the viewpoint of the arguments presented here, but physically equivalent. We also review their current status in the light of the new findings of our work.

**First law of species thermodynamics**

*In the limit $T \rightarrow \Lambda_{sp}$ the entropy and energy must satisfy*

$$S \sim N_{sp}, \quad E \sim N_{sp} \Lambda_{\text{sp}}. \tag{108}$$

We refer to these as the *constitutive relations* of species thermodynamics. As argued in section 3, one of the main goals of this work has been to *derive* these relations from the standard thermodynamics of a system of species in the appropriate limit, without the need to make any extra assumptions. We have found that a system of KK-like or string-like species in thermal equilibrium in a box of size $L \simeq 1/T$, as required for the system not to collapse into a black hole, is indeed described by the relations above in the limit $T \rightarrow \Lambda_{\text{sp}}$, as well as recovers the usual volume scaling of the energy and entropy in the limit where $T \gg 1/L$. This relation between energy and entropy is derived from the first law of (standard) thermodynamics, for a system in which there the $d$-dimensional momentum degrees of freedom are frozen (or subleading), as we have shown is the case in the aforementioned limit.

Once these constitutive relations have been derived, and given that our original system follows the standard laws of thermodynamics, one could be tempted to conclude that the laws of species thermodynamics are automatically satisfied. The only subtlety related to this reasoning has to do with the laws of thermodynamics that involve time evolution, like the second law that we revisit below. The key points in the formulation of species thermodynamics can be encapsulated in the identification of the *appropriate* constitutive relations, and the identification of time in the usual laws of thermodynamics with geodesic distance in moduli spaces in the laws of species thermodynamics. Our progress in this work mainly concerns the first aspect, but not explicitly the second. It is thus guaranteed that the laws of thermodynamics not involving time evolution are automatically satisfied in their species thermodynamics counterparts, but the ones related to time evolution require extra assumptions, on which we comment below.

## Second law of species thermodynamics

*As one moves towards the boundary of moduli space, the species scale must not increase, and the entropy must not decrease.*

We consider two points in moduli space $\phi_i$, $\phi_f$, and a boundary point $\phi_B$, such that the distance in moduli space between them satisfies $\Delta(\phi_i, \phi_B) > \Delta(\phi_f, \phi_B)$. Assuming the Distance Conjecture [9], the mass scale (in Planck units) of an infinite tower of states, $m_t(\phi)$, must decrease as one moves towards the boundary. Hence, for sufficiently large $\phi_i$ and $\phi_f$, $m_t(\phi_i) > m_t(\phi_f)$, and the second law can then be stated as

$$\Lambda_{\text{sp}}(\phi_i) \geq \Lambda_{\text{sp}}(\phi_f), \quad S(\phi_i) \leq S(\phi_f), \tag{109}$$

where $\Lambda_{\text{sp}}$ is again measured in Planck units. Given the constitutive relations of Species thermodynamics it is enough to show that $\Lambda_{\text{sp}}$ decreases as one decreases the mass. For a tower of states with mass scale $m_t$, the number of species below the temperature $T$ must increase as one decreases the mass scale, or equivalently, it must increase as one increases the temperature. The second law is then equivalent to the requirement $\partial N_T / \partial T > 0$. In other words, this law follows directly from the simple requirement that the spectra from which the species tower is composed can be considered as an *appropriate* tower, as described in section 3.3.

Let us then use this fact to give a complementary argument in favor of the second law of species thermodynamics by relating it to the standard laws of thermodynamics for our system of species. First, let us remark that the entropy of the systems of species that we consider (in the limit in which the $d$-dimensional momenta are effectively frozen) is exclusively controlled by the ratio $T/m_t$, as discussed in sections 3.1 to 3.3. In this sense, decreasing the mass scale of the tower with respect to a fixed temperature is equivalent to increasing the temperature in units of $m_t$, and within this identification, the requirement $\partial N_T / \partial T > 0$ is morally equivalent to the Distance Conjecture. With this in mind, we consider a point $\phi_i$ in the bulk of moduli space, and a point $\phi_B$ near the boundary, with associated mass scales for the towers that become light given by $m_i = m_t(\phi_i)$ and $m_B = m_t(\phi_B) \ll m_i$, respectively. If we now take the relevant thermodynamic systems at both points (i.e. the boxes of species in thermal equilibrium with frozen $d$-dimensional momenta), characterized by temperatures $T_i$ and $T_B$ near the corresponding values of the species scale at each point, we have $\frac{T_B}{m_B} \gg \frac{T_i}{m_i}$, $S_B \gg S_i$ and $\frac{E_B}{m_B} \gg \frac{E_i}{m_i}$. Then, in order to analyze the two in a thermodynamic setup we need to compare them at the same point in moduli space. To this purpose, we can move the system at $\phi_i$ towards the boundary point $\phi_B$ along a constant entropy trajectory, such that the temperature in units of $m_t(\phi)$, and therefore the entropy, remain constant. Once both systems are at the same point in moduli space, we can then put them in thermal contact and depict this in terms of a thermodynamic system at temperature $T_i$ in contact with a large reservoir at temperature $T_B$ (note that we can consider the system at $T_B$ as a reservoir because its energy is arbitrarily larger than that of the system that we have transported from $\phi_i$, and we can thus neglect the loss of energy and temperature as long as the heat transfer does not break this assumption). The second law of standard thermodynamics then implies that heat will flow from the hotter system to the cooler system and that the entropy of the smaller system will increase. Then, we can let this process go up to a point where the temperature of our system is greater than $T_i$ but still arbitrarily lower than $T_B$ (in order to avoid breaking the reservoir condition). Our initial thermal system has then an entropy greater than $S_i$. If we then transport it back along a constant entropy trajectory, we see that it reaches the maximum possible temperature (i.e. $T \simeq \Lambda_{\text{sp}}(\phi)$) at a point $\phi_f$ which is closer to the boundary than $\phi_i$, since we know that the entropy has increased and thus the ratio $m_t(\phi_f)/\Lambda_{\text{sp}}(\phi_f)$ must be smaller than at $\phi_i$, which as we argued above can be associated to the point $\phi_f$ being closer to the boundary. Let us remark that we have made the key assumption that one can freely move in moduli space along constant

entropy lines, or more generally that systems with the same entropy are thermodynamically equivalent. We will revisit this assumption in Chapter 5.

**Third law of species thermodynamics**

*It is impossible by any physical process to reduce the species temperature $T \simeq \Lambda_{sp}$ to zero by a finite sequence of operations.*

This is a direct application of the third law of thermodynamics. For an spectrum satisfying the constitutive relations of species thermodynamics one has (in $d$-dimensional Planck units)

$$E \simeq \Lambda_{\mathrm{sp}}^{3-d} . \tag{110}$$

We can then note that (for $d > 3$), the limit of vanishing species scale requires an infinite amount of energy. In fact, as commented in [21], if the only two towers allowed are KK and string towers, the limit of $\Lambda_{\mathrm{sp}}$ going to 0 corresponds to either a decompactification limit, or the limit of a weakly coupled string becoming tensionless. Both of these processes require infinite energy, and the third law follows from this.

# 4 Entropies in lower and higher dimensions

In this section we briefly highlight some important aspects of black holes and black branes in the presence of $p$ extra dimensions. In particular, we focus on the correspondence and differences between what we denote the *EFT black hole*, which for $R_{\mathrm{BH}} < r$ we identify as the black brane solution wrapping the extra dimensions of size $r$, and the *higher-dimensional black hole* solution, which for $R_{\mathrm{BH}} < r$ is a fully localized solution in the higher-dimensional theory, namely a $(d + p)$-dimensional spherical black hole with $R_{\mathrm{BH}} < r$ in the compact and non-compact dimensions

## 4.1 The EFT entropy

Recall that for a given an EFT configuration at finite temperature, $T < \Lambda_{UV}$, in a box of volume $L^{d-1}$, and with a finite ($\mathcal{O}(1)$) light species, the total entropy takes the form

$$S_{\mathrm{EFT},d} \simeq T^{d-1} L^{d-1} . \tag{111}$$

We have also seen in section 3 that in the presence of $N_T$ species contributing to the thermal ensemble at temperature $T$ the entropy (in the thermodynamic limit $T \gg L$) is given by

$$S \simeq N_T T^{d-1} L^{d-1} . \tag{112}$$

Then, for a KK tower originating from the presence of $p$ extra dimensions, which we assume to be isotropic for simplicity, and of typical size $r = 1/m_{\mathrm{t}}$, we can rewrite the entropy as

$$S_{\mathrm{EFT},p,D} \simeq T^{d+p-1} r^p L^{d-1} . \tag{113}$$

Note that this is nothing but the entropy of a $D = d+p$ dimensional EFT in a box of size $L$ along the $d$ non-compact dimensions, and $r$ along the $p$ compact ones, such that the it completely wraps the compact dimensions. From this perspective, the scale at which (111) and (113) meet $T \simeq m_{\mathrm{t}}$, is simply the energy scale at which the EFT starts to perceive the modes in the tower, and they can be in thermal equilibrium.

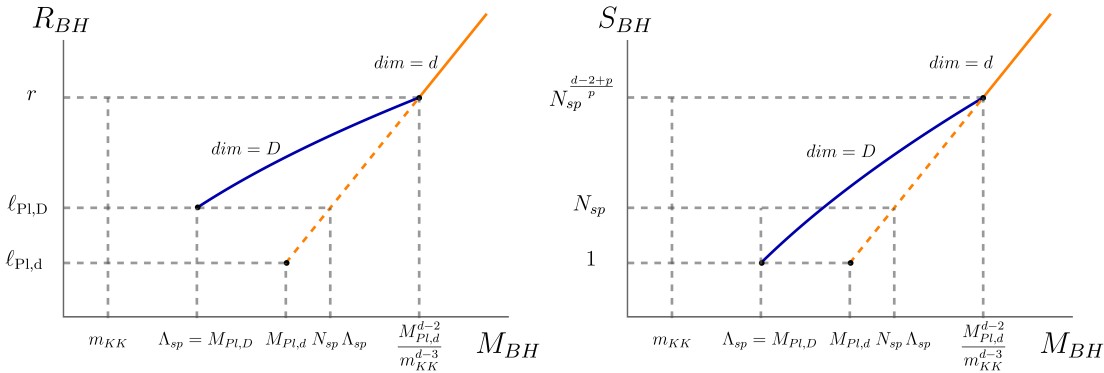

Figure 2: Horizon radius and entropy versus black hole mass. The orange line corresponds to a lower-dimensional black hole, while the blue line corresponds to the higher-dimensional one. The dashed, orange line corresponds to a black brane solution wrapping the extra dimensions. At equal mass, the higher dimensional black hole is more entropic.

## 4.2 Black hole vs. black brane entropy

A $d$-dimensional black hole of radius $R_{\text{BH}}$ has entropy

$$S_{\text{BH},d} \simeq R_{\text{BH}}^{d-2} M_{\text{Pl},d}^{d-2} = M_{\text{BH}}^{\frac{d-2}{d-3}} M_{\text{Pl},d}^{-\frac{d-2}{d-3}} . \tag{114}$$

Since it is a purely gravitational object, it is expected that as one decreases its radius the description near scales $L \simeq r$ must change in order to reflect the presence of extra dimensions. For $R_{\text{BH}} < r$ one can first consider a $D$ dimensional black hole localized in the extra dimensions, namely one with

$$S_{\text{BH},D} \simeq R_{\text{BH}}^{D-2} M_{\text{Pl},D}^{D-2} = M_{\text{BH}}^{\frac{D-2}{D-3}} r^{\frac{p}{D-3}} M_{\text{Pl},d}^{-\frac{d-2}{D-3}} . \tag{115}$$

On the other hand, a black object with radius $R_{\text{BH}}$ along the $d$ non-compact and wrapping the whole internal manifold can be seen as a black brane solution [63], and the total surface gains a contribution from the internal volume, $r^p$. The entropy in this case reads

$$S_{\text{BB},p,D} \simeq R_{\text{BB}}^{d-2} r^p M_{\text{Pl},D}^{D-2} = M_{\text{BB}}^{\frac{d-2}{d-3}} M_{\text{Pl},d}^{-\frac{d-2}{d-3}} . \tag{116}$$

As first noticed by Gregory and Laflamme [64], the black brane is unstable for $R_{\text{BB}} < r$, as the higher dimensional black hole with the same mass, $M_{\text{BH}} = M_{\text{BB}}$, has a greater entropy, as displayed in Fig. 2. Using the relation between Planck masses

$$M_{\text{Pl},d}^{d-2} = r^p M_{\text{Pl},D}^{D-2} , \tag{117}$$

one can see that the lower-dimensional black hole solution and the black $p$-brane one have parametrically the same entropy. We can compare EFT and black hole entropies in the specific limit of $R_{\text{BH}} \simeq 1/T \simeq 1/\Lambda_{UV} \simeq 1/M_{\text{Pl},D}$ and we see that the black brane (and the lower dimensional black hole) can be identified with the EFT that sees the presence of the species of the tower. In contrast, a higher dimensional black hole that does not wrap the internal directions contains no information about the tower. This is to be expected, as the fully localized solution along the extra dimensions can only see the localized modes along the extra dimensions, as opposed to the full tower of KK modes including all momentum modes with $1/r \leq T$. Furthermore, from an EFT perspective, it is natural to remain agnostic about the details of the extra dimensions. From the black solution point of view, this idea can be interpreted as maximizing the entropy associated to the extra dimensions by wrapping them, and thus being sensitive to all states in the KK tower up to temperatures $T = 1/R_{\text{BH}} \geq 1/r$.

# 5 The black hole - tower correspondence: Black hole entropy from non-gravitational thermodynamics

This section is devoted to the analysis of the entropy of Schwarzschild black holes in EFTs from the microstate counting of non-gravitational systems that can be obtained from following the adiabats of such black holes towards infinite distance limits. The following discussion can be understood as a generalization of the celebrated Black Hole - String correspondence[17] [30–32] (see also [62, 65–67] for recent discussions on the topic), which accounted for the entropy of general Schwarzschild black holes (up numerical prefactors, but crucially matching the area dependence) by the microscopic counting of the microstates associated to a free string in the limit of vanishing string coupling. We begin with a review of the Black Hole - String correspondence, following mainly [30, 65], and then proceed to explain the generalization arising in the presence of general infinite distance limits.

## 5.1 The black hole - string correspondence

In its crudest incarnation, the key idea behind the Black Hole - String correspondence is to consider a Schwarzschild black hole solution at some finite string coupling, and follow it adiabatically as the string coupling is decreased. Following the black hole along this adiabatic process (through which the entropy stays constant) as the string coupling is decreased, it turns out that at some point the gravitational interaction becomes too weak for the black hole to remain a black hole. At that point, it transitions to a long, highly tangled string[18] that one can then also follow along the corresponding adiabatic trajectory while decreasing the string coupling until a point in which entropy can actually be computed as the entropy of the free string. Thus, the black hole entropy of any Schwarzschild black hole (at finite string coupling) can actually be understood from microstate counting of a free string. We now proceed to describe this process more quantitatively, and comment on the limitations of the picture along the way.

First, let us recall some useful facts about black holes and strings. To begin with, the Planck and string lengths are related via the $d$-dimensional dilaton, $g_{s,d}$, as follows

$$\ell_{\mathrm{Pl,d}}^{d-2} = g_{s,d}^2 \, \ell_{\mathrm{str}}^{d-2} \,. \tag{118}$$

Furthermore, the relation between the mass and radius of a black hole in $d$-dimensions is given by

$$M_{\mathrm{BH}} \sim \frac{R_{\mathrm{BH}}^{d-3}}{\ell_{\mathrm{Pl,d}}^{d-2}} \sim \frac{R_{\mathrm{BH}}^{d-3}}{g_{s,d}^2 \, \ell_{\mathrm{str}}^{d-2}} \,, \tag{119}$$

and its entropy, which is proportional to the area in Planck units, takes the form

$$S_{\mathrm{BH}} \sim \left( \frac{R_{\mathrm{BH}}}{\ell_{\mathrm{Pl,d}}} \right)^{d-2} \sim \left( M_{\mathrm{BH}} \ell_{\mathrm{Pl,d}} \right)^{\frac{d-2}{d-3}} \sim g_{s,d}^{\frac{2}{d-3}} \left( M_{\mathrm{BH}} \ell_{\mathrm{str}} \right)^{\frac{d-2}{d-3}} \,, \tag{120}$$

where we have expressed all quantities both in Planck and string units for later convenience using (118).

---

[17]In this work we refer to the *correspondence*, as opposed to the *transition*, to emphasize the fact that we are not studying the details of the transition as a dynamical process, but focusing instead on the correspondence of the entropies of the black hole system and the tower system to obtain a microscopic interpretation of the entropy of the former in terms of the microstate counting of the latter.

[18]In principle it could transition to a gas of strings, but the main contribution to the entropy for such gas turns out to be a single, long string, as discussed in section 3.

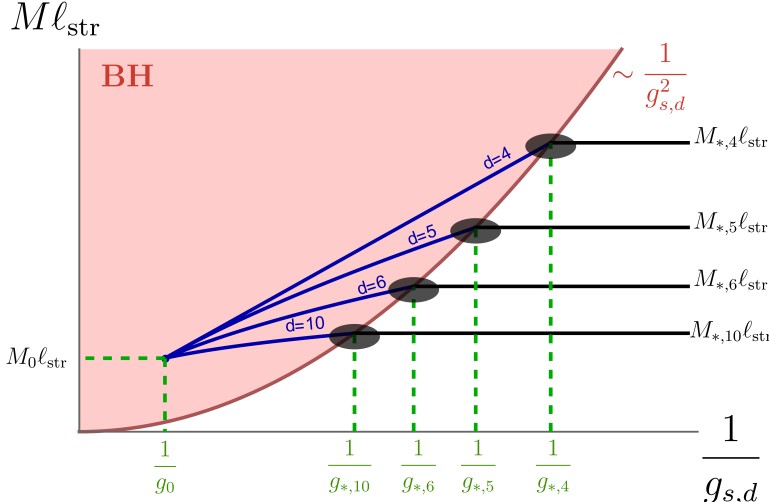

Figure 3: Constant entropy trajectories in the $M - g_{s,d}^{-1}$ plane for black hole solutions (in blue) and for string solutions (in black) for different number of dimensions, starting from a given mass $M_0$ at a given string coupling $g_0$. The red line indicates the transition region between the black hole and the string solutions, characterized by $R_{\text{BH}} \sim \ell_{\text{str}}$.

Additionally, if we consider a long, highly excited, free string of length $L_{\text{str}}$, its mass and entropy are given by (see e.g. [65])[19]

$$M_{\text{str}} \sim \frac{L_{\text{str}}}{\ell_{\text{str}}^2}, \qquad S_{\text{str}} \sim \frac{L_{\text{str}}}{\ell_{\text{str}}} \sim M_{\text{str}} \ell_{\text{str}}. \tag{121}$$

Now, let us start by considering a black hole with mass $M_0$ (in string units) for a value of the string coupling given by $g_0$. The constant entropy lines for the black hole solution in the $M - g_{s,d}^{-1}$ plane (see Fig. 3) are given by

$$M_{\text{BH}} \ell_{\text{str}} \sim \frac{g_0^{\frac{2}{d-2}} M_0 \ell_{\text{str}}}{g_{s,d}^{\frac{2}{d-2}}}. \tag{122}$$

Furthermore, as we go towards $g_{s,d} \to 0$, the radius of the black hole solution, measured in string units, decreases. Hence, we can start with a very large, semiclassical black hole, at $g_0$, but as we adiabatically decrease the string coupling it will always reach a point at which $R_{\text{BH}} \sim \ell_{\text{str}}$, where the gravitational interaction would become so weak that it would not be able to keep the black hole together and the transition to a string (or gas of strings) should occur. The transition region defined by $R_{\text{BH}} \sim \ell_{\text{str}}$ takes the following form in the $M - g_{s,d}^{-1}$ plane

$$M \ell_{\text{str}} \sim \frac{1}{g_{s,d}^2}, \tag{123}$$

and the values of the aforementioned relevant quantities at the transition region are

$$g_{*,d} \sim \left( g_0^{\frac{2}{d-2}} M_0 \ell_{\text{str}} \right)^{-\frac{d-2}{2(d-3)}}, \qquad M_* \ell_{\text{str}} \sim \frac{1}{g_{*,d}^2}. \tag{124}$$

---

[19]A quick and intuitive way to understand these formulae is to consider the string as given by a random walk of $N$ steps in a $d$-dimensional lattice of size given by $\ell_{\text{str}}$. The length and mass of all such configurations would be given by adding the $N$ steps, each corresponding to $\ell_{\text{str}}$, producing $L_{\text{str}} \sim N \ell_{\text{str}}$ and $m_{\text{str}} \sim N/\ell_{\text{str}}$. Furthermore, the number of possible configurations grows exponentially with $N$, thus giving and entropy $S_{\text{str}} \sim N$.

In fact, the entropy (which is constant along the adiabatic trajectory) can be written as

$$S \sim \frac{1}{g_{*,d}^2} \sim M_{*,d}\, \ell_{\mathrm{str}}, \tag{125}$$

matching precisely the entropy of a string of mass $m_{\mathrm{str}} \sim M_{*,d}$ (c.f. eq. (121) ). Thus, at the transition point, namely when the radius of the black hole becomes of the order of the string length, the mass of the black hole coincides with the mass of the free string that would account for the same entropy. Even if the details regarding the exact transition are not fully clear, one can still follow the potential string solution that matches the entropy along its corresponding adiabatic trajectory towards arbitrarily weak coupling, where the microscopic understanding of the entropy of the free string is fully reliable. The constant entropy lines in the $M - g_{s,d}^{-1}$ are the horizontal lines

$$M_{\mathrm{str}}\, \ell_{\mathrm{str}} \sim M_{*,d}\, \ell_{\mathrm{str}}. \tag{126}$$

Therefore, for any (large) semiclassical black hole at fixed string coupling, following the adiabatic trajectory as the string coupling is reduced, there will always be a transition to a string that we can then follow to arbitrarily weak coupling to count the degrees of freedom that account for the entropy of the starting black hole. Let us remark that this is possible only because for every black hole solution, the constant entropy trajectories always intersect the transition region $R_{\mathrm{BH}} \sim \ell_{\mathrm{str}}$, as otherwise we could not use the previous argument. Notice that this is at the core of our discussion in section 3.3 regarding *appropriate* towers. This is the case since in this language a tower of string excitations can equivalently be identified as *appropriate* due to the fact that it gives rise to the right entropy and energy in the limit in which it should collapse to a black hole of size $R_{\mathrm{BH}} \sim \ell_{\mathrm{str}}$, and moreover there is a solution giving the right behavior for any value of $g_{s,d} \ll 1$.

Several remarks are in order. First, this method can successfully recover the area dependence of the entropy in Planck units, but not the order one prefactors, due to the fact that we do not have a clear picture about the details of the transition. This is to be contrasted with detailed computations including order one factors, such as the one in [68], where supersymmetry and extremality allow for a detailed counting. However, note that this dependence is enough for our purposes, as we are interested in understanding the area dependence of the entropy in the presence of towers of species. In this regard, one could expect gravitational effects to be more relevant [31, 32], causing a departure from the free string picture near the transition point, for the cases in which the transition takes place at only moderately small coupling. However, one would expect the approximation to be more and more precise if one starts with a sufficiently large black hole at $g_0$ such that the transition happens at arbitrarily small $g_{*,d}$, and thus the area dependence of the entropy of an infinite number of black holes (those with masses $M \geq M_0$ at $g_0$), would also be more precisely accounted by the free string computation. In any event, independently of numerical, order one prefactors that might depend on the details of the transition (represented by the black bubble in Fig. 3), the main message is that one can still follow the adiabat towards arbitrarily small $g_{s,d}$ to perform the counting of the states in the free string solution.

## 5.2 Decompactification limit and KK towers

Let us now try to generalize the aforementioned process to the case in which instead of exploring a weak string coupling point, we probe a decompactification limit of the EFT. We focus here on a situation in which the decompactification limit is not accompanied by an arbitrarily weak string coupling limit, such that the KK tower is asymptotically lighter than any of the string excitations, and the higher dimensional species scale is asympotically the same as the

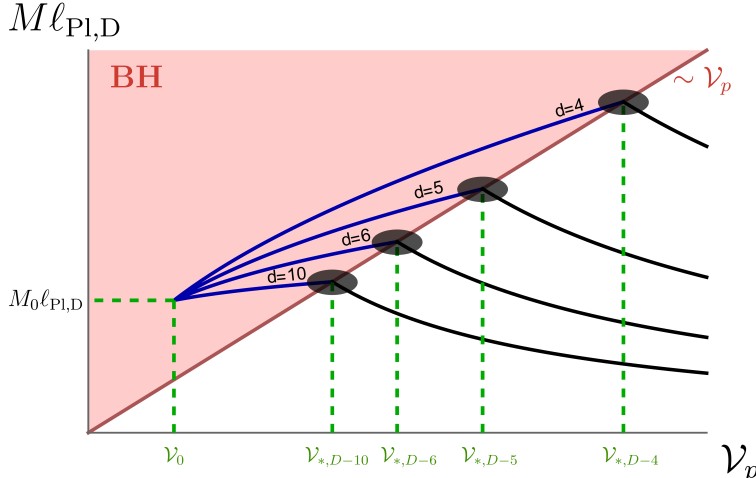

Figure 4: Constant entropy trajectories in the $M - \mathcal{V}_p$ plane for black hole solutions (in blue) for different number of dimensions, starting from a given mass $M_0$ at a given value for the internal volume, $\mathcal{V}_0$. The red line indicates the transition region between the black hole and the tower, characterized by $R_{\mathrm{BH}} \sim \ell_{\mathrm{Pl,D}}$.

higher dimensional Planck mass. From the top-down construction, this includes both the decompactification to 10-dimensional string theory at fixed $g_s$ or to 11-dimensional M-theory. Following the same logic as for the black hole - string correspondence, we start with a large, semiclassical black bole of mass $M_0$ in a $d$-dimensional EFT, now obtained from compactification of a $D$-dimensional theory (i.e. $D = d + p$). We denote by $\mathcal{V}_p$ the volume of the $p$ internal dimensions (measured in higher dimensional Planck units) and consider our starting point to be given by $\mathcal{V}_0^{1/p} \ell_{\mathrm{Pl,D}} \ll R_{\mathrm{BH}}$. The higher and lower dimensional Planck lengths are related as

$$\ell_{\mathrm{Pl,d}}^{d-2} = \frac{\ell_{\mathrm{Pl,D}}^{d-2}}{\mathcal{V}_p}, \tag{127}$$

and the black hole and mass and entropy, given in eqs. (119)-(120), read as follows in $D$-dimensional Planck units

$$M_{\mathrm{BH}} \sim \frac{R_{\mathrm{BH}}^{d-3}}{\ell_{\mathrm{Pl,D}}^{d-2}} \mathcal{V}_p, \qquad S_{\mathrm{BH}} \sim \left( \frac{M_{\mathrm{BH}}^{d-2} \ell_{\mathrm{Pl,D}}^{d-2}}{\mathcal{V}_p} \right)^{\frac{1}{d-3}}. \tag{128}$$

Analogously to the black hole - string transition case discussed above, we can now try to follow the constant entropy lines as we increase $\mathcal{V}_p$, effectively reducing the intensity of $d$-dimensional gravity, c.f. (127). Such constant entropy lines in the $M - \mathcal{V}_p$ plane (see Fig. 4) are given by

$$M \ell_{\mathrm{Pl,D}} \sim \frac{M_0 \ell_{\mathrm{Pl,D}}}{\mathcal{V}_0^{\frac{1}{d-2}}} \mathcal{V}_p^{\frac{1}{d-2}}, \tag{129}$$

and it can be checked that as we follow them along the direction of increasing $\mathcal{V}_p$, the radius of the corresponding black hole becomes smaller and smaller in $D$-dimensional Planck units.

There are two interesting points along this trajectory. First, since we are decreasing the black hole radius at the same time as we make the internal volume larger, we will reach a point at which both sizes are comparable, namely $R_{\mathrm{BH}} \sim \mathcal{V}_p^{1/p} \ell_{\mathrm{Pl,D}}$. From that point on, there is an ambiguity on the way see our black hole solution in the higher dimensional theory, as

discussed in section 4. In particular, we can continue with a $D$-dimensional black hole localized in the extra dimensions, or a black brane solution whose horizon is spherical in $d$-dimension but wraps the internal dimensions completely. In this section we are interested in the second possibility, since this is the one in which the tower associated to the decompactification limit, namely the KK modes, actually plays a role in accounting for the entropy. In the former case, we could not continue along a constant entropy line while still reducing the radius of the horizon without including some extra weak coupling point, such as an additional weak string coupling limit, effectively recovering the picture presented by the black hole string transition. Furthermore, let us remark that choosing to follow the black brane trajectory is not in conflict with the idea that the localized black hole of equal mass is more entropic when $R_{\mathrm{BH}} \sim m_{\mathrm{KK}}^{-1}$, as recently highlighted in the Swampland context in [69]. At that point, we simply focus on the less entropic solution because it still wraps the extra dimensions, and it is thus the only one that allows us to probe the limit $R_{\mathrm{BH}} \to \ell_{\mathrm{Pl,D}}$ in $d$ dimensions while following constant entropy lines. Since we are not following the dynamical evolution of the black hole solutions, this is allowed.

Let us then focus on the $d$-dimensional black hole that actually wraps the whole $p$-dimensional manifold even when it effectively becomes larger than the horizon, which from the higher dimensional point of view corresponds to the black brane. Incidentally, this morally looks like the more sensible thing to do from the lower dimensional EFT perspective, since being agnostic about the details coming from the internal dimensions should correspond to a maximum uncertainty about their details, which in this case nicely fits with the idea of the horizon hiding them. Thus, from this point one could consider a sort of black hole - black $p$-brane transition. In any event, this is still a semiclassical object in the theory and it can actually be checked that the area of its horizon, which accounts for the entropy, is still correctly described by eq. (128) (c.f. eq. (116)), and thus the constant entropy lines are also given by (129).

The second special point along this constant entropy trajectory turns out to be the most relevant one for our argument. This is the point at which the $d$-dimensional black hole horizon reaches its minimum allowed size, namely $R_{\mathrm{BH}} \sim \ell_{\mathrm{Pl,D}}$ (assuming the $D$-dimensional UV cutoff is sufficiently well approximated by the $D$-dimensional Planck scale, as is the case unless an arbitrarily high number of species different from the KK ones were asymptotically lighter than this scale). In the $M - \mathcal{V}_p$ plane this transition region is defined by

$$M\ell_{\mathrm{Pl,D}} \sim \mathcal{V}_p \,. \tag{130}$$

Once again, we see that for any $d > 3$ this always intersects the constant entropy lines in the black hole region. This intersection occurs for the following values of the internal volume and the mass

$$\mathcal{V}_{*,p} \sim \left( \frac{M_0^{d-2}\ell_{\mathrm{Pl,D}}^{d-2}}{\mathcal{V}_0} \right)^{\frac{1}{d-3}} , \qquad M_*\ell_{\mathrm{Pl,D}} \sim \mathcal{V}_{*,p} \,. \tag{131}$$

Furthermore, we can recall that the number of species associated to the KK-modes corresponding to decompactification of $p$-dimensions is precisely given by

$$N_{\mathrm{sp}} \simeq \left( \frac{\ell_{\mathrm{Pl,D}}}{\ell_{\mathrm{Pl,d}}} \right)^{d-2} = \mathcal{V}_p \,, \tag{132}$$

where we have used that the species scale and the higher dimensional Planck scale coincide. More interestingly, the entropy can be rewritten as the number of species corresponding to the intersection point

$$S \sim \mathcal{V}_{*,p} \sim N_* \,. \tag{133}$$

This is precisely the species entropy, and as we have discussed in previous section, it can be understood as the entropy associated to $N_*$ *free* species frozen in a box of size $L \sim \ell_{\mathrm{Pl,D}}$ at a value of the modulus $\mathcal{V}_{*,p}$. Thus, following the constant entropy trajectories for values larger than $\mathcal{V}_{*,p}$ the black hole should transition to something else that near the transition region can be thought of as a system of $N_*$ species in a box. Once again, independently of the details of the transition, we can still follow the constant entropy lines towards larger $\mathcal{V}_p$, and also avoiding gravitational collapse. Hence, we approach sufficiently weak $d$-dimensional gravity, and perform a counting of the entropy in the free theory near that point, where arbitrarily good control can be achieved, to account for the entropy of the original black hole of mass $M_0$ at $\mathcal{V}_0$. In particular, for such a box of particles, we can now play with two different control parameters in order to compute the entropy, namely the size of the box and the temperature. In the black hole case these are related, so we can only tune one of them freely. Notice that in the case of the free string, for $T \simeq T_{\mathrm{H}} \simeq m_{\mathrm{str}}$, which is the temperature near which the transition region is probed, we have $T = (\partial S / \partial M)^{-1} \sim \ell_{\mathrm{str}}^{-1}$, so following the free string and including its excitation modes we effectively reduced the control parameters to one. For the case of the KK tower, we can explore two interesting limits after the transition region to compute the entropy in the $\mathcal{V}_p \to \infty$ limit. These correspond precisely to two of the limits discussed in section 3.1, namely the one in which $T \simeq 1/L$ (dubbed frozen momentum limit) where the entropy is dominated by the number of species below $T$, and the one where $T \gg 1/L$ but with $T \lesssim m_{\mathrm{t}}$ (dubbed thermodynamic limit), such that the effective number of species is order one and the entropy is dominated by the momentum configurations in the box associated to the massless modes.

The former case can be achieved by the scaling

$$T \simeq \frac{1}{L} \sim \frac{N_*^{1/p}}{\ell_{\mathrm{Pl,D}} \mathcal{V}_p^{1/p}} \,. \tag{134}$$

In such case, we have $S \simeq N_T \simeq N_*$, such that as we lower the temperature while exploring the limit $\mathcal{V}_p \to \infty$, we keep the number of excited species constant and also make the size of the box larger in a way that the momentum modes of the species in $d$-dimensions are still frozen. The entropy is thus accounted for by the number of active species below $T$. Crucially, $T \ll \ell_{\mathrm{Pl,D}}^{-1} \ll \ell_{\mathrm{Pl,d}}^{-1}$ and also we can stay arbitrarily below the gravitational collapse bound (18), such that neglecting the gravitational interactions is justified in this limit, as was the case in section 3. Thus, we can account for the entropy of the initial black hole in the EFT by computing the entropy associated to the free theory in the infinite distance limit. The entropy is therefore given by that of a tower of KK modes, c.f. eq. (77),

$$S \sim \left( M \ell_{\mathrm{Pl,D}} \mathcal{V}_p^{1/p} \right)^{\frac{p}{p+1}} \,, \tag{135}$$

where the tower mass $M$ is given by its total energy $E$. Constant entropy trajectories then follow

$$M \ell_{\mathrm{Pl,D}} \sim \frac{M_0 \, \ell_{\mathrm{Pl,D}} \mathcal{V}_0^{1/p}}{\mathcal{V}_p^{1/p}} \,, \tag{136}$$

such that at the transition point $M \ell_{\mathrm{Pl,D}} = \mathcal{V}_p$ one obtains $S = S_{\mathrm{BH}}$, $L = R_{\mathrm{BH}}$.

The later limit can be obtained by the scaling

$$T \simeq \frac{N_*^{\frac{1}{d-1}}}{L} \sim \frac{1}{\ell_{\mathrm{Pl,D}} \mathcal{V}_p^{1/p}} \,. \tag{137}$$

In this case, at large $\mathcal{V}_p$ the temperature is below the lightest massive states in the tower, and thus the massless species dominate the entropy. In this case, however, their $d$-dimensional momentum states can be excited, due to larger size of the box with respect to $T$, i.e. $L \simeq N_*^{\frac{1}{d-1}}/T$,

and it is this entropy that accounts for the one of the original black hole. In any event, let us remark that even if in the limit $\mathcal{V}_p \gg \mathcal{V}_{*,p}$ one can tune $T$ to recover any of these two behaviours, it is still necessary to connect this with the line $T \simeq 1/L$, namely (134), close to the transition region in order to match the entropy.

## 5.3 The black hole - tower correspondence

Having studied the two kinds of towers associated to infinite distance limits, namely string oscillator modes and KK-like towers, in the context of a correspondence that allows us to account for the entropy of a semiclassical black hole at finite effective gravitational coupling from microstate counting in the free limit, we are now in a position to formulate this in terms of a general *Black Hole - Tower Correspondence*, that includes both cases. Following the previous cases, the general logic is to start with some black hole solution at finite value of the effective gravitational coupling, and study the constant entropy trajectories as we make that coupling smaller. This process reduces the radius of the corresponding black hole solution (in the right units, which we will identify with $\Lambda_{sp}$), and at some point it reaches the minimum possible radius for a black hole in the EFT. Around that region, we expect some transition to take place, in which a high number of particles (the ones giving rise to the tower that relates the effective gravitational coupling and the species scale) take over the black hole description. Independently of the details of the actual transition, which occupy a great part of this paper (and also previous works related to species thermodynamics [26–28]), we must remark that by continuing along our adiabatic trajectory towards arbitrarily weak effective gravitational coupling we can account for the entropy of the original black hole from the usual thermodynamics of the corresponding gas of species in the gravity decoupling limit, in close analogy to the way in which the free string could account for the entropy of the original black hole in the black hole - string correspondence. The general picture is then as follows. We consider a $d$-dimensional

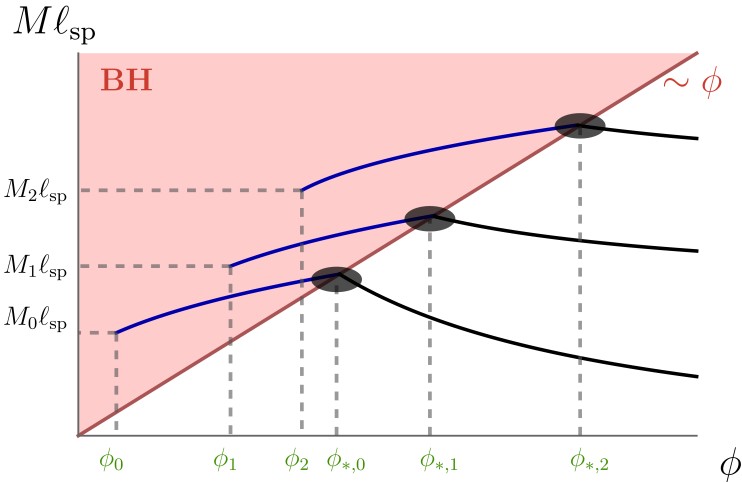

Figure 5: Constant entropy trajectories in the $M - \phi$ plane for different black hole solutions (in blue) for different initial conditions starting from a given mass $M_n$ at a given value for the modulus, $\phi_n$. The red line indicates the transition region between the black hole and the tower, characterized by $R_{BH} \sim \ell_{sp}$. We see that independently of initial conditions all configurations can be moved towards the transition region by varying the modulus $\phi$. Here we are assuming arbitrary values of $p$.

EFT, where the $d$-dimensional Planck length is related to the species scale length as

$$\ell_{\text{Pl,d}}^{d-2} = \frac{\ell_{\text{sp}}^{d-2}}{\phi}, \tag{138}$$

with $\phi$ the modulus controlling the effective gravitational coupling in such a way that the latter decouples in the $\phi \to \infty$ limit. One can then extrapolate the expressions for the black hole and tower entropies as

$$S_{\text{BH}} \sim \frac{1}{\phi^{d-3}} \left( M_{\text{BH}}^{d-2} \ell_{\text{sp}}^{d-2} \right)^{\frac{1}{d-3}}, \qquad S_{\text{tower}} \sim \phi^{\frac{1}{p+1}} \left( M \ell_{\text{sp}} \right)^{\frac{p}{p+1}}, \tag{139}$$

where we are using the parameterization of the tower in which $p$ encodes the number of dimensions that are being decompactified and recovers the stringy case for $p \to \infty$. In Fig. 5 we depict the black hole - tower transition in the $M - \phi$ plane for different initial conditions. The core idea is that, for any initial conditions, and independently of the value of $p$, one can always vary the modulus in order to reach the transition region. It is well-known that $\ell_{\text{sp}}$ captures the higher dimensional Planck mass or the string length associated to the infinite distance limit that we probe as $\phi \to \infty$, which recovers the cases described above upon identifying $\phi = \mathcal{V}_p$, $g_s^{-2}$, respectively. It is thus straightforward to see that both the black hole - string correspondence and the black hole - KK tower correspondence described above can be reformulated in terms of $\ell_{\text{sp}}$ and $\phi$. The general idea is that indeed the transition from the black hole to the tower should take place in the region $R_{\text{BH}} \sim \ell_{\text{sp}}$, where we have

$$S \sim \phi_* \sim N_* \sim M_* \ell_{\text{sp}}, \tag{140}$$

and the dependence on the details of the tower is encoded via $N_* \sim \phi_*$, which can be computed from different towers. After the transition region, we can follow our constant entropy trajectories as we decouple the effective gravitational coupling as $\phi \to \infty$ and compute the entropy of the initial black hole from the thermodynamics of the tower. The two most interesting cases in which we can compute the entropy in the free theory are

$$T \simeq \frac{1}{L} \sim \frac{N_*^{1/p}}{\ell_{\text{sp}} \phi^{1/p}}, \qquad \text{and} \qquad T \simeq \frac{N_*^{\frac{1}{d-1}}}{L} \sim \frac{1}{\ell_{\text{sp}} \phi^{1/p}}, \tag{141}$$

where we remark once again that this freedom to chose the scaling of the temperature to include a higher or lower number of species at temperature $T$ is only available for $\phi \gg \phi_*$, but we need to match the trajectory to the former, with $T \simeq 1/L$, near the correspondence region to keep the entropy constant. The first case is such that the entropy is accounted for by the number of species below $T$, whereas in the second it is accounted by the $d$-dimensional momentum configurations of the massless modes. It can be seen that the $p \to \infty$ case degenerates in the sense that both cases give rise to $T \sim \ell_{\text{sp}}^{-1}$. In any case, it is important to remark that in these cases it is justified to neglect the gravitational interactions in the limit, since

$$T \ll \ell_{\text{sp}} \ll \ell_{\text{Pl,d}} \qquad \text{(for finite } p), \tag{142}$$

$$T \sim \ell_{\text{sp}} \ll \ell_{\text{Pl,d}} \qquad \text{(for } p \to \infty). \tag{143}$$

Thus, in the first case, both the $d$-dimensional and the $D$-dimensional gravitational interactions are negligible, and in the second the gravitational interaction is negligible and the stringy effects are taken into account by the free string even for $T \sim \ell_{\text{sp}}$.

It can also be checked that, in the limit, the energy of the configurations is below the gravitational collapse threshold for a configuration of size $L$. For the case $T \simeq 1/L$, the requirement that $E(L) \leq M_{\mathrm{BH}}(R_{\mathrm{BH}} = L)$ can be written as

$$\frac{N_T}{L} \lesssim \left(\frac{L}{\ell_{\mathrm{Pl,d}}}\right)^{d-2} \frac{1}{L}, \tag{144}$$

where we have used eqs.(47) and (119). This can be brought to the form

$$N_* \leq \phi, \tag{145}$$

which is parametrically saturated precisely at the transition point $\phi \sim \phi_* \sim N_*$, and fulfilled for any larger value of $\phi$, as expected. Similarly, the CEB is also fulfilled for any $\phi \geq \phi_*$ and is saturated at the correspondence point.

Finally, to connect with the usual conventions in the swampland literature, where quantities are meaningfully measured in $d$-dimensional Planck units, let us remark that at the correspondence point, namely when $R_{\mathrm{BH}} \sim \Lambda_{\mathrm{sp}}^{-1}$, the mass of the black hole is

$$\frac{M_*}{M_{\mathrm{Pl},d}} \simeq \left(\frac{\Lambda_{\mathrm{sp}}}{M_{\mathrm{Pl},d}}\right)^{3-d}. \tag{146}$$

so that we can always associate the mass scale $\Lambda_{\mathrm{sp}}^{3-d}$ as the one given by the mass of the smallest black hole in the EFT (in Planck units), whose corresponding temperature is $T \simeq \Lambda_{\mathrm{sp}}$.

## 6 Conclusions

The goal of this work was twofold. First, we have shown that the constitutive relations of species thermodynamics [26], namely $S_{\mathrm{sp}} = E_{\mathrm{sp}}/\Lambda_{\mathrm{sp}} = N_{\mathrm{sp}}$, can be *derived* from standard thermodynamics of a system of species in thermal equilibrium. This can also be motivated by previous analysis about the maximum entropy of a system in the presence of species before gravitational collapse takes place, and the interplay between the Covariant Entropy Bound and the species scale [37, 52]. Second, investigating which kinds of towers can give rise to such constitutive relations in the limit $T \to \Lambda_{\mathrm{sp}}$, understood as a key feature of Quantum Gravity that allows for an interpolation between the usual volume dependence of the entropy in field theory and the area behaviour for black holes. This can be motivated and comprehended from the bigger picture of allowing for a black hole - tower correspondence, in analogy with the black hole - string correspondence [30–32].

We have started by reviewing how the Covariant Entropy Bound, together with the gravitational collapse bound, allows for a system to (asymptotically) approach the maximum entropy in the presence of gravity by tuning the control parameters to effectively freeze the $d$-dimensional momentum degrees of freedom. In other words, we have highlighted how, if one tries to get as close as possible to the saturation of the Covariant Entropy Bound (i.e. $S \sim A$) without previously collapsing the thermodynamic system of species into a black hole, one is led to set $T \simeq 1/L$ (or $T \sim T_{\mathrm{H}}$ for a string tower), so effectively freezing the momentum modes. Then, in the limit $T \to \Lambda_{\mathrm{sp}}$ both bounds coincide [37] and one recovers $S \sim N_{\mathrm{sp}} \sim (M_{\mathrm{Pl},d}/\Lambda_{\mathrm{sp}})^{d-2}$. This means that it is justified to take the thermodynamic system and try to obtain the area scaling of the entropy in the aforementioned limit from the standard thermodynamics of a system of species (see also [25] for an alternative explanation of the area dependence of the related *entropy of species* from saturation of unitarity). Let us also mention that the fact that the EFT kinematic degrees of freedom freeze as one approaches the maximum

temperature (while avoiding gravitational collapse), whereas they dominate the entropy and energy at low temperatures (in particular for $T \ll m_\mathrm{t}$), is immediately reminiscent of some ideas behind the Emergence Proposal [3, 70–73] (see also [17, 19, 22, 74–80]), which in its strong version states that light particles in the IR have no kinetic terms in the UV. It would be very interesting to explore this connection in the future.

With this in mind, we have analyzed a system of KK-like or string-like species, corresponding to polynomial or exponential degeneracies in the spectrum, respectively, in thermodynamic equilibrium at temperature $T$. In our analysis, we neglect interactions, which is justified as long as we avoid gravitational collapse and study weakly couple regimes, or equivalently, large number of species. As a first consistency check, we find that both the energy and the entropy scale with the volume as $S \sim E/T \sim (LT)^{d-1}$ for low temperatures (compared with $\Lambda_\mathrm{sp}$), as expected in the usual field theory limit, in which the available momentum states dominate the entropy and the energy. In contrast, we derive that for high-enough temperatures, such that a large number of species $N_T$ contributes to the canonical partition function, and ensuring that the system does not collapse gravitationally, a scaling of the entropy and energy that goes like $S \sim E/T \sim N_T$ is recovered. We also show that this converges to $N_T \to N_\mathrm{sp}$ as $T \to \Lambda_\mathrm{sp}$. From a purely thermodynamical analysis in the canonical ensemble we have then found an interpolation between the usual volume scaling of the energy and entropy at low temperatures, where momentum states dominate, and their scaling with the number of active species at high temperatures, where the system only makes sense if the momentum modes are frozen to avoid gravitational collapse. This latter limit is precisely the one we show to converge to the constitutive relations of species thermodynamics in the $T \to \Lambda_\mathrm{sp}$ limit, where the entropy scales with the area.

Let us also briefly remark that the canonical ensemble analysis provides an interesting perspective on the species scale in the presence of a tower of string-like excitations. In particular, from a microcanonical analysis, as well as from perturbative definitions of the species scale, it is known that some subtleties arise from the apparent mismatch between the identification of the species scale with the string mass, and the fact that no states in the tower lie below such scale. The reason being that this naively implies $N_\mathrm{sp} \sim 1$, which is inconsistent with the fact that $\Lambda_\mathrm{sp} \sim m_\mathrm{str} \ll M_{\mathrm{Pl},d}$ from the definition (1). A self-consistent counting, though, is known to produce an extra multiplicative logarithmic factor [16, 17] that still does not match the higher-curvature computations [18, 23, 24] (see also [19, 27, 81] for some related discussions on the topic), raising a puzzle. In contrast, interpreting the species scale as the limiting temperature in the canonical ensemble, which is arguably a correct definition of UV cutoff, seems to provide the following consistent picture: For low enough (polynomial-like) degeneracies, only states with masses below $T$ can contribute, so that only those with masses below $\Lambda_\mathrm{sp}$ enter its computation, whereas for high enough degeneracies (exponential-like), also states above $T$ can contribute significantly to the partition function, providing a self-consistent picture in which the tower of string excitations above $m_\mathrm{str} \sim T_\mathrm{H}$ can contribute and set $\Lambda_\mathrm{sp} \sim m_\mathrm{str}$. Let us remark this turns out to be a key feature to take into account in e.g. cosmological applications of string configurations near the Hagedorn temperature, like the ones recently studied in [82]).

Having recovered the constitutive relations of species thermodynamics from our first-principles thermodynamic analysis of KK-like and string-like towers, we have then investigated the question of whether different kinds of towers can also give rise to such form of the entropy and energy, so as to to recover the black hole scaling in the $T \to \Lambda_\mathrm{sp}$ limit. First, we have discarded any kind of super-exponentially degenerate spectrum, since they give rise to a divergent partition function for any $T > 0$. These can be complementary analyzed in the microcanonical ensemble, but have also been found to be inconsistent with species thermodynamics [27, 28]. One could also argue that it is sensible to introduce an extra hard cutoff in the

masses of the canonical partition function, so as to make it convergent. As we have seen, this cutoff prescription can at most be thought of as an effective way of parameterizing an exponentially degenerate tower, morally similar to the $p \to \infty$ limit in the polynomial case. Second, we have found that towers with fixed mass or sub-polynomial degeneracy cannot provide the right scaling for the energy and the entropy at the same time, and are thus not *appropriate* towers. The exception being when their mass scale satisfies $m_{\mathrm{t}} \simeq \Lambda_{\mathrm{sp}}$, which also resembles an effective parameterization of a string-like tower. Thus, we have provided new evidence for the Emergent String Conjecture [10], in agreement with recent bottom-up arguments from the microcanonical analysis of species thermodynamics [27] and general properties of scattering amplitudes [29].

Finally, we have interpreted our results in the bigger picture of a black hole - tower correspondence. Let us mention that a correspondence between species and minimal black holes was also suggested in [27] from a different perspective, but our proposal extends beyond that and allows to set the correspondence between black holes of any size and towers of species at arbitrarily weak gravitational coupling, formulating a correspondence which is on equal footing with the original argument that led to the proposal of the black hole - string correspondence [30, 31]. In fact, revisiting such original reasoning, it is transparent that the existence of such correspondence between black holes and free strings can be understood in terms of the presence of a (non-gravitational) system whose entropy and energy are the correct ones to match those of the black hole when the latter becomes as small as possible, which in our language means $R_{\mathrm{BH}} \simeq \Lambda_{\mathrm{sp}}^{-1}$. This correspondence is thus established when a black hole solution is followed along constant entropy lines towards weak gravitational coupling, by varying a modulus (originally the string coupling) up to the point when its constant entropy curve intercepts that of the free string with equal mass. Besides, this should not only happen for one particular black hole solution, but for all of them, as long as they possess sufficiently high entropy for the parametric analysis to be meaningful. We have shown that this can then be generalized in the presence of a system of species that is able to reproduce the right scaling of the entropy and the energy in the limit $T \to \Lambda_{\mathrm{sp}}$, that is, a tower that is able to produce the constitutive relations of species thermodynamics, $S \sim E/\Lambda_{\mathrm{sp}} \sim N_{\mathrm{sp}}$, in said limit. Thus, all our previous arguments about the towers that we dubbed *appropriate* can be rephrased in terms of towers that do not obstruct a black hole - tower correspondence. In particular, we can now formulate this correspondence not only for emergent string limits, as originally proposed, but also for KK-like limits, where the KK species account for the entropy of the EFT black hole that wraps the corresponding extra dimensions.[20] Hence, from this perspective, the towers that are allowed in Quantum Gravity should be the ones that can account for the (asymptotic) entropy of black hole solutions as they are followed along constant entropy lines towards arbitrarily weak gravitational coupling. Let us remark that our analysis here is not meant to explain the details of the transition, but instead we see it as a first step (and a necessary condition) towards properly establishing the detailed correspondence between towers of species and black holes. That is, the way to think about this is as the equivalent of the argument in [30] for the black hole - string correspondence, but several subtleties regarding the details of the transitions, such as those originally discussed in [31, 32] (and more recently [62, 65–67]) would also need to be considered to understand the full picture in detail.

---

[20]Incidentally, let us remark that this latter claim is consistent with the idea of a more entropic black hole solution in decompactification limits when one probes sizes of order $R_{\mathrm{BH}} \sim m_{\mathrm{KK}}^{-1}$, as recently highlighted in the Swampland context in [69]. At that point, we focus on a less entropic solution for a given mass, namely the one that still wraps the extra dimensions, since this is the one that allows us to probe the species scale while following constant entropy lines. Since we are not focusing on the dynamical evolution of the black hole solutions, following these is justified.

## Acknowledgments

We would like to thank Ivano Basile, José Calderón-Infante, Alberto Castellano, Niccolò Cribiori, Nicolás Kovensky, Yixuan Li and Carmine Montella for useful discussions.

**Funding information** The work of D.L. is supported by the Origins Excellence Cluster and by the German-Israel-Project (DIP) on Holography and the Swampland.

## A  Single-particle canonical partition function

We devote this appendix to clarify the approximations used in the main text for the single-particle canonical partition function of a relativistic particle of mass $m_n$ in a box of size $L$ in $(d-1)$-spatial dimensions, at a temperature $T \leq \Lambda$. First, we recall such partition function, given by eq. (25), which we recall here for simplicity

$$\mathcal{Z}_{1,n} = \sum_{p_\alpha} e^{-E_{n,p_\alpha}/T}, \qquad E_{n,p_\alpha}^2 = m_n^2 + \sum_{\alpha=1}^{d-1} p_\alpha^2, \tag{A.1}$$

with $p_\alpha = k_\alpha/L$ (where $k_\alpha$ are non-vanishing integers). For $L^{-1} \ll T$, the number of points in the lattice determined by the allowed $k_\alpha$'s which gives the leading contribution to the sum is large and we can approximate it by an integral

$$\mathcal{Z}_{1,n} \simeq L^{d-1} \int_{1/L}^{\infty} dp \; p^{d-2} \exp\left(-\frac{\sqrt{m_n^2 + p^2}}{T}\right), \tag{A.2}$$

where we have defined the modulus of the spatial momentum $p^2 = \sum_{\alpha=1}^{d-1} p_\alpha^2$, as well as neglected factors of $2\pi$ in the definition of the volume of phase space, since we will eventually neglect multiplicative order one factors and it is not meaningful to keep track of them at this point. The change of variables $x = \sqrt{\frac{m_n^2 + p^2}{T^2}}$ brings this integral to the form

$$\mathcal{Z}_{1,n} \simeq (TL)^{d-1} \int_{\sqrt{\frac{m_n^2}{T^2} + \frac{1}{(TL)^2}}}^{\infty} dx \; x \left(x^2 - \frac{m_n^2}{T^2}\right)^{\frac{d-3}{2}} e^{-x}, \tag{A.3}$$

which for odd $d$ can be performed analytically because the quantity inside the parenthesis becomes a polynomial in $x$. Solving the integral analyticallly in those cases and substituting the integration limits one obtains a product including the $(TL)^{d-1}$, and exponential functions of $1/(TL)$, and $m_n/T$ multiplying polynomials on the same variables. These expression can be expanded in powers of $m_n/T$ and the following results are obtained

$$\begin{aligned} \mathcal{Z}_{1,n} &\simeq (TL)^{d-1}, && \text{for } T \geq m_n, \\ \mathcal{Z}_{1,n} &\simeq e^{-m_n/T} P_q\left(\frac{m_n}{T}\right)(TL)^{d-1}, && \text{for } T \ll m_n. \end{aligned} \tag{A.4}$$

These results are valid for any $1/L \leq T \leq \Lambda$, as long as the integral gives a good approximation for the sum, and only include the leading term, which is in general multiplied by functions of $1/(TL)$ and $\Lambda/T$, which are extremely stable in the range of values under consideration and can be approximated by a constant. Furthermore, $P_q(m_n/T)$ represents a polynomial in $m_n/T$

with degree $q \leq (d-1)/2$. As an example, let us consider in detail the result of the $d = 5$ case, where the integral takes the form

$$\mathcal{Z}_{1,n}^{(5d)} \simeq (T L)^{d-1} \left[ e^{-x} \left\{ \left( \frac{m_n^2}{T^2} - 6 \right) x + \frac{m_n^2}{T^2} - x^3 - 3x^2 - 6 \right\} \right]_{x=\sqrt{\frac{m_n^2}{T^2} + \frac{1}{(TL)^2}}}^{x=\infty} , \tag{A.5}$$

and we can substitute the limits explicitly and expand about different values for $m_n/T$, keeping in mind that $L^{-1} \leq T \leq \Lambda$:

$$\mathcal{Z}_{1,n}^{(5d)} \simeq (TL)^4 \left\{ e^{-\frac{1}{TL}} \left[ \frac{1}{L^3 T^3} + \frac{3}{L^2 T^2} + \frac{6}{LT} + 6 \right] + \mathcal{O}\left( \frac{m_n}{T} \right)^2 \right\} , \qquad \text{for } T \gg m_n , \tag{A.6}$$

$$\mathcal{Z}_{1,n}^{(5d)} \simeq (TL)^4 \left\{ e^{-\frac{\sqrt{1+(TL)^2}}{TL}} \left( \frac{6\sqrt{(TL)^2 + 1}}{(TL)} + \frac{3}{(TL)^2} + \frac{\sqrt{(TL)^2 + 1}}{(TL)^3} + 8 \right) \right\} , \qquad \text{for } T \simeq m_n , \tag{A.7}$$

$$\mathcal{Z}_{1,n}^{(5d)} \simeq e^{-\frac{m_n}{T}} (TL)^4 \left\{ 2\frac{m_n^2}{T^2} + \frac{m_n}{T} \left[ \frac{1}{TL} + 6 \right] + \frac{3}{(TL)^2} + 6 + \mathcal{O}\left( \frac{T}{m_n} \right) \right\} , \qquad \text{for } T \ll m_n . \tag{A.8}$$

Let us start by analyzing eq. (A.6). In this case we have the factor $(T L)^{d-1}$ and, using the definition of the incomplete gamma function $\Gamma(d-1, x)$, the function multiplying it can be rewritten as

$$\mathcal{Z}_{1,n}^{(5d)} \simeq \Gamma\left( 4, \frac{1}{TL} \right) (TL)^4 , \quad \text{for } T \gg m_n . \tag{A.9}$$

This prefactor is a multiplicative function bounded by $\Gamma(d-1) \geq \Gamma(d-1, 1/(TL)) \geq \Gamma(d-1, 1)$, which are saturted in the limits $T \gg L^{-1}$ and $T \simeq L^{-1}$, respectively. Therefore, we obtain the behaviour described in (A.4)

For the regime $T \simeq m_n$ it can be checked that the factor multiplying $(T L)^4$ in eq. (A.7) is of order one in all possible limits for the parameters (i.e. $T \geq L^{-1}$), and this is compatible with (A.4)

Finally, we can see that for the case $T \ll m_n$ the factor $(T L)^4$ is exponentially suppressed in eq. (A.8), as in (A.3). Furthermore, the leading term in the extra multiplicative correction is indeed a monomial in $m_n/T$ with degree $q = 2$, which matches (A.3) for $d = 5$

It can be checked that all other odd dimensional cases yield similar results upon performing the integral analytically. The even dimensional cases can also be analyzed upon expanding the integrand for the different relevant limits, and yield the same leading approximations, summarized in eq. (A.4).

To end this section, we focus now on the case in which the number of momentum states that give the leading contribution to the single particle partition function is small, such that we can try to estimate the value of the sum in eq. (A.1) directly, without the need to use the integral approximation. This corresponds to the limit $T \simeq L^{-1}$. In this case, if $m_n \lesssim T$, there will be $\mathcal{O}(1)$ contributions to the partition function which are not heavily exponentially suppressed and are of $\mathcal{O}(1)$ each. On the other hand, if $m_n \gg T$ all contributions will be heavily exponentially suppressed. Thus, this behaviour is also captured by eq. (A.4), since for the former case, $(T L)^{d-1} \sim \mathcal{O}(1)$, and for the latter we obtain the right exponentially suppressed behavior.

**Fluctuations in the canonical ensemble**

In this subsection we collect some useful formulae regarding the fluctuations in equilibrium configurations at temperature $T$. In general, for a given distribution function, we can define the (squared) fluctuation of a magnitude $Q$ as [33,34]

$$(\Delta Q)^2 = \langle Q^2 \rangle - \langle Q \rangle^2 . \tag{A.10}$$

This characterizes the width of the distribution function around its average value, $\langle Q \rangle$. It is in particular useful to consider the adimensional quantity $\Delta Q / \langle Q \rangle$, which we denote *relative fluctuation*, since it gives the relative measure of the width of the distribution normalized by the average value. If this relative fluctuation is small, the relative width of the distribution is narrow. In the particular case in which the magnitude $Q$ represents the energy, whenever we have a relative fluctuation that approaches zero, the distribution in energies approaches a delta and we recover the microcanonical ensemble.

For the energy, its average value can be computed from the partition function as

$$\langle E \rangle = T^2 \frac{\partial \log(\mathcal{Z})}{\partial T}, \tag{A.11}$$

and using this it can be seen that

$$(\Delta E)^2 = \langle E^2 \rangle - \langle E \rangle^2 = T^2 \frac{\partial \langle E \rangle}{\partial T}. \tag{A.12}$$

## B  Towers in the microcanonical ensemble

We compliment the discussion on towers in the main text by showing equivalent results using the microcanonical ensemble. Here we work at finite temperature, but one should always take the limit $T \simeq \Lambda_{\mathrm{sp}}$ at the end.

### B.1  Towers with polynomial degeneracy

We begin by considering towers with the following mass spectrum and degeneracy

$$m_n = n\, m_{\mathrm{t}}, \qquad d_n = n^{p-1}, \tag{B.1}$$

corresponding to a tower of KK-like modes from the compactification of $p$ isotropic dimensions. The number of species in this case is given by $N_T \simeq N^p/p$, and the entropy takes the form

$$S = N_T \frac{p+1}{p} + N_T \log\left(\frac{M}{N_T^{\frac{p+1}{p}}}\right). \tag{B.2}$$

The entropy is then maximized in terms of $N_T$ for

$$M = N_T^{\frac{p+1}{p}}. \tag{B.3}$$

This configuration is precisely the one which reproduces the expected results of species thermodynamics, in which each state contributes exactly once to the total energy

$$E = M\, m_{\mathrm{t}} \simeq \sum_{n=1}^{N} m_n d_n \simeq m_{\mathrm{t}} N^{p+1} \simeq m_{\mathrm{t}} N_T^{\frac{p+1}{p}}. \tag{B.4}$$

The entropy and temperature then take the form

$$S = N_T\left(\frac{p+1}{p}\right), \tag{B.5}$$

$$T = m_{\mathrm{t}} N_T^{1/p}, \tag{B.6}$$

and we recover eq.s (14) and (11).

### B.2 Towers with fixed mass

We now consider the following spectrum

$$m_n = n\,m_{\rm t}\,, \qquad d_n = \delta_{n,1}N\,. \tag{B.7}$$

This corresponds to modes accumulating around some mass $m_{\rm t}$ with degeneracy $N$. The number of available species is simply $N_T = N$, with entropy

$$S = N + N\log\left(\frac{M}{N}\right) = N + N\log\left(\frac{T}{m_{\rm t}}\right), \tag{B.8}$$

where we have used eq. (53) to express $N$ in terms of the temperature.

Maximizing the entropy requires

$$M = N\,, \tag{B.9}$$

such that

$$S = N\,, \tag{B.10}$$

$$T = m_{\rm t}\,. \tag{B.11}$$

In contrast to the previous case, this suggests that such a tower is only in an equilibrium configuration for $T = m_{\rm t}$, as one cannot relate the temperature with the number of available modes $N_T$.

### B.3 Towers with exponential degeneracy

We finally consider string-like towers with the following mass spectrum

$$m_n = \sqrt{n}\,m_{\rm t}\,, \qquad d_n = e^{\lambda\sqrt{n}}\,. \tag{B.12}$$

The number of modes in the tower is given in terms of the level as

$$N_T \simeq \frac{2}{\lambda^2}e^{\lambda\sqrt{N}}\lambda\sqrt{N}\,. \tag{B.13}$$

The entropy is

$$S \simeq N_T + N_T\log\left(\frac{M}{N_T\log N_T}\right), \tag{B.14}$$

the configuration maximizing the entropy is

$$M \simeq N_T\log N_T\,. \tag{B.15}$$

The equilibrium entropy and temperature for string towers are then

$$S \simeq N_T\,, \tag{B.16}$$

$$\Lambda_{\rm sp} \simeq m_{\rm t}\log N_{\rm sp}\,. \tag{B.17}$$

This corresponds directly to the number of states kinematically available with masses below $T$. We can thus notice that the microcanonical counting can be suited for degeneracies which are up to exponential, as the results match with those of the canonical partition function.

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
