# Peer review of "On the Origin of Species Thermodynamics and the Black Hole - Tower Correspondence"

_SciPost Physics, doi:SciPost Phys. 18, 083 (2025)_

## Round 1 · Referee Report · Anonymous (Referee 1) · 2025-1-15

Report
The authors carefully analyze configurations of species in a box of size $L$ by separating into several regimes for the temperature $T$. The rules of species thermodynamics are recovered in the limit where $L^{-1}=T$ approaches the species scale $\Lambda_{sp}$, with the entropy scaling as the area of the box. Away from this limit the entropy scales in the usual way as the volume of the box, or a mixture between these two cases.
In turn, these rules of species thermodynamics are subsequently applied to configurations of species given by towers of particles. It is found that the only consistent towers must have a degeneracy that grows at least polynomially and at most exponentially. This provides important evidence for the emergent string conjecture, as these degeneracy rates correspond to Kaluza-Klein (KK) towers and string excitations respectively.
Finally, the authors propose a black hole-tower correspondence, which is a generalization of the well-established black hole-string correspondence. This extends the black hole-string correspondence, where usually the string coupling is varied, to KK-like limits with varying radii of the extra dimensions. They do not investigate the details of this transition, but rather leave it as a compelling direction for future research.
The paper is well-written and I recommend it for publication in SciPost. It easily fulfills the journal expectations listed by the authors, as it connects several research areas and concepts (swampland, covariant entropy bound, black hole-string correspondence), and opens up interesting follow-up opportunities for research (species thermodynamics, black hole-tower correspondence).
Recommendation
Publish (easily meets expectations and criteria for this Journal; among top 50%)
Author: Alvaro Herraez on 2025-02-04 [id 5184]
(in reply to Report 2 on 2025-01-17)First of all we would like to thank the referee for their careful report, positive feedback and for highlighting several improvements to be made. Let us now address the different points one by one:
1) This sentence simply serves as an introduction for the rest of the paragraph, where all the references (from [12] to [24]) are cited in a more detailed way, explaining the context in which they are relevant. We plan to rephrase this in a future version to make it clearer.
2) The laws of species thermodynamics are explicitly presented in section 3.4, as well as our derivation of them. We agree with the referee, though, that a brief explanation in the introduction would be convenient for the reader, and plan to add it below eq. (1.3), where species thermodynamics is mentioned for the first time, and the original references are provided.
3) The key observation for the formulation of the Black Hole - Tower correspondence, in analogy to the Black Hole String correspondence, is the fact that the latter can be formulated in the language of species. Upon identifying the species length with the string length, $\ell_{\mathrm{sp}}\sim \ell_{\mathrm{str}}$, the relation between the Planck scale and the species scale is controlled by $\ell_{\mathrm{Pl},d}^{d-2}\sim g_{s,d}^2 \ell_{\mathrm{sp}}^{d-2}$ (see eq. (5.1) in the manuscript), and the correspondence takes place at very weak gravitational coupling, namely when $g_{s,d}\ll 1$ (since then $\ell_{\mathrm{Pl},d}/\ell_{\mathrm{sp}} \ll 1$), at the mass $M_{\ast} \ell_{{\mathrm{str}}}\sim 1/g_{s,d}^2\sim S$ (see eq. (5.6) ). Precisely at that point, the entropies and masses coincide. Similarly, this can be performed for a KK tower associated to $p$-dimensions, upon the right identification of the species scale, $\ell_{\mathrm{sp}}\sim \ell_{\mathrm{Pl},d+p}$, and the modulus controling the ratio between the latter and the $d$-dimensional Planck mass, which is nothing but the volume of the internal dimensions, i.e., $\ell_{\mathrm{Pl,}d}^{d-2}\sim \frac{\ell_{\mathrm{Pl,}d+p}^{d-2}}{\mathcal{V}_p}$(see eq. (5.10)). Thus, the very weak (d-dimensional) gravitational coupling happens for $\mathcal{V}_p \gg 1$, and the transition takes place at $M_\ast \ell_{\mathrm{sp}}\sim \mathcal{V}_p \sim S$, where the entropies and masses also coincide. The general idea is explained in section 5.3, where identifying $\phi$ in eq. (5.21) with either $g_{s,d}^{-2}$ or $\mathcal{V}_p$ one recovers the the correspondence points of both cases. A detailed explanation of all this is the content of section 5.
4)We thank the referee for spotting this typo, we will correct it in a future version.
5) The definition of $P_q$ is indeed presented in the appendix, but we completely agree with the referee that it would be clearer if also stated below (3.6). We plan to add it there in a future version.
6) As explained in the text, the motivation for these two conditions is twofold. First, they are fulfillied in all examples we know in top-down constructions, namely for KK towers and for towers of weakly coupled string oscilators. Second, these conditions are the minimal set of conditions that recover the thermodynamic properties of schwarzschild BHs for temperatures in the limit $T\to \Lambda_{\mathrm{sp}}$, and towers that do not fulfill them would never be able to match the energy and entropy of black holes in such limit, which is the key property of “appropriate towers”, as explained in section 3. Additionally, we will restate the second condition in a clearer way in a future version, as suggested by the referee.
7) We will clarify that $\Delta$ refers to the distance above (3.87) in a future version. We will also replace "SDC"->"Distance Conjecture" in page 32 for clarity.
8) We thank the referee for spotting the typo, we will correct it.
9) The subtlety lies in the fact that the identification $\Lambda_{\mathrm{sp}}\sim m_{\mathrm{str}}$ naively implies $N_{\mathrm{sp}}\sim 1$, and this is inconsistent with $\Lambda_{\mathrm{sp}}\ll M_{\mathrm{Pl}}$ according to the definition of species scale (c.f. eq. (1.1) in the manuscript). As explained in the text, a self-consistent counting can then be performed but yields extra logarithmig factors that do not match the species scale identified from higher-curvature corrections, raising a puzzle. The canonical ensemble interpretation allows for a consistent resolution of such puzzle, since it allows for an arbitrarily high-number of species to contribute, even if their masses are above $\Lambda_{\mathrm{sp}}$. This is the case only if their degeneracy can compesate the Boltzman suppression factor, which is precisely the case for a string tower, since it possesses an exponential degeneracy. We plan to state this in a clearer way in the corresponding paragraph in the conclusions.
10) The object that we study for the black hole-tower transition in decompactification limits is the one that wraps the extra dimensions. Once its horizon in the $d$ non-compact dimensions becomes $R_{BH}\lesssim m_{\mathrm{KK}}^{-1}$, for a given mass there might be a more entropic object —even though this is not fully clear according to recent discussions on the Gregory-Laflamme instability in string theory, as also pointed by the referee in his Question 3, namely the fully localized spherical black hole in $D$-dimensions. However, as dicussed in section 5.2.—the pargraph below eq. (5.12)—and footnote 20 (see also the reply to question 3 below), this does not play a crucial role for our argument, since for us it is enought o have meta-stable black brane solution. Furthremore, these are the only ones that allow us to probe the species scale in the non-compact dimensions while still following constant entropy lines as the extra dimensions decompactify.
Questions: 1) This is indeed the next natural step in establishing the black hole-tower correspondence, as mentioned in the manuscript, but it is beyond the scope of the present work— we are currently working on it as a separate project.
2) Adiabaticity is indeed a crucial part for the general argument that motivates the black hole-tower correspondence. In section 5 we refer to “constant entropy lines” and “adiabatic trajectories” indistinguishably, and all the diagrams depicted there follow such adiabatic trajectories.
3) In the context of decompactification limits, the Gregory-Laflamme instability can definitely play an important role when studying dynamical transitions of black hole solutions. However, for the arguments in this paper (and as explained in section 5.2.—the pargraph below eq. (5.12)—and footnote 20) it does not play a crucial role, given that we focus on the space of (meta-)stable black hole solutions and follow constant entropy lines, as opossed to studying the dynamical decays of such objects. Additionally, the references pointed out by the referee were not included simply because they appeared after the last version of our manuscript was uploaded to arXiv, but we agree they could be relevant for future extensions of our work.

---

## Round 1 · Referee Report · Anonymous (Referee 2) · 2025-1-17

Report
The paper discusses several aspects of the role that many species play in the definition of EFTs and within the context of the Swampland program. The authors discuss "species thermodynamics", derive the constitutive relations from standard thermodynamics, and see where the species cutoff comes in the context of the covariant entropy bound. They argue that there are only two possible cutoffs that are consistent with their analysis, namely the KK decompactification scale and the string scale, and as evidence, they present an analysis that gets rid of all other possibilities. Finally, they generalize the black hole-string correspondence principle to include the KK scale as well, transforming the principle into that of black hole-tower correspondence.
The paper is interesting, and it has a lot of consistency checks. The paper certainly meets this journal's criteria, and therefore I recommend it be published.
However, it would be beneficial for the readers if the authors would address the following questions and remarks:
1) The last sentence on page 1 mentions several arguments for why the species scale should be the final EFT cutoff, but no elaboration is provided nor a reference. Even though these might be familiar arguments, at least a reference would be in place. 2) The concept of "species thermodynamics" is never defined in the paper. At least the introduction should have the definition of it so that the reader is clear on what is being discussed. In particular, the difference between standard thermodynamics and species thermodynamics is not explained well: if we are simply introducing a new scale, why is there a need for a different type of thermodynamics etc. Please elaborate on this. 3) I am a little bit confused about where the black hole-tower correspondence point lies. Usually for the bh-string correspondence, the string coupling is inversely proportional to the entropy; is the same being done here? Please elaborate. 4) There is a typo in equation 2.10: it should be 2-d, not d-2. 5) In equation 3.6, it is not defined what is meant by P_q (perhaps it is in the appendix, but it should be readable from the main text). 6) On page 22, when the authors discuss the conditions of an appropriate tower, first, it is not clear why these two conditions are put and not some others (seems arbitrary), and since it is the crux on which the constitutive relations are derived from, the authors should better and in a clearer way motivate this notion of appropriate towers; second, the second condition is almost unreadable-please rewrite it less confusingly. 7) Above equation 3.87, \Delta is not defined (I assume it is the distance, but please be consistent). Likewise, on the same page 32, SDC is not defined (again, I can assume what it is, but still). 8) Typo in the second sentence in the Conclusions section (twice derived deriving). 9) In the second paragraph on page 48, the authors discuss some subtleties but they do not explicitly say what are they about; please provide an explanation and/or a reference (why is there a mismatch, how does this new paper resolve it explicitly, etc.). I understand that an explanation was intended for that paragraph, but it was, unfortunately, written in an unclear way. 10) Did I understand correctly from your footnote 20 that the black holes of size m^{-1}_{KK} are never dominant thermodynamically (either in the microcanonical or canonical)? This seems to be different compared to the standard bh-string correspondence picture in which a bh transitions into a stringy solution. Could you comment more on this, if possible?
Questions: 1) In the string thermodynamics framework, one deals with the thermal scalar formalism, as defined by Atick and Witten in the 80s. Can you comment on a possible "species thermal scalar" or if such a thing even makes sense? If not, please explain why not. 2) A crucial part of the black hole-string correspondence is the need for adiabaticity as one goes from one description to another. Do you need to impose a similar notion here? Please comment. 3) Does the new correspondence play a role in the Gregory-Laflamme instability? If so, in what way? There were several recent papers on this topic within the bh-string correspondence (https://inspirehep.net/literature/2844283, https://inspirehep.net/literature/2865831, https://inspirehep.net/literature/2851290). Do you see a connection with their approach?
Recommendation
Publish (easily meets expectations and criteria for this Journal; among top 50%)

---

## Round 2 · Author Response

We thank both referees for their careful reports. We have addressed the points raised in the Referee Report#2 in the author replies, and have adapted the manuscript to reflect their recommendations (as detailed in the list of changes below).

---

## Round 2 · List of Changes

• The sentence at the end of page 1 — after eq. (1.1)— was reformulated slightly to make the point more clear.

  • The sentence at the end of page 2 — after eq. (1.3)— was modified to include a brief definition of species thermodynamics, together with a reference to section 3.4, where they are dicussed in more detail.

  • A typo noticed by the referee was fixed in eq. (2.10), the same typo was also fixed in eq. (1.3).

  • The explicit definition of $P_q$ was added below eq. (3.6) for clarity.

  • On page 22, after eq. (3.41), the second condition for appropriate towers was reformulated to make it more clear.

-$\Delta$ was explicitly introduced as the "distance in moduli space" above (3.87), when it first appears.

  • The acronym "SDC" was replaced by "Distance Conjecture" in the text.

  • Typo corrected in the first paragraph of Section 6: "can be derived deriving" -> "can be derived".

  • We re-wrote different parts of the fourth paragraph in the Conclussions to explain more clearly what the subtleties are when identifying the species scale with the string scale, as well as to explain how the canonical ensemble analysis that we present offers a new perspective on it.

---

## Editorial Decision

published